



# Additional carbon inputs to reach a 4 per 1000 objective in Europe: feasibility and projected impacts of climate change based on Century simulations of long-term arable experiments

Elisa Bruni[1], Bertrand Guenet[1,2], Yuanyuan Huang[3], Hugues Clivot[4,5,] Iñigo Virto[6], Roberta Farina[7], Thomas Kätterer[8], Philippe Ciais[1], Manuel Martin[9], Claire Chenu[10]

[1]Laboratoire des Sciences du Climat et de l'Environnement, LSCE/IPSL, CEA-CNRS-UVSQ, Université Paris-Saclay, F-91191 Gif-sur-Yvette, France

[2]LG-ENS (Laboratoire de géologie) - CNRS UMR 8538 - École normale supérieure, PSL University - IPSL, 75005 Paris France

[3]CSIRO Oceans and Atmosphere, Aspendale 3195, Australia

[4]Université de Lorraine, INRAE, LAE, 68000, Colmar, France

[5]Université de Reims Champagne Ardenne, INRAE, FARE, UMR A 614, 51097 Reims, France

[6]Departamento de Ciencias. IS-FOOD, Universidad Pública de Navarra, 31009 Pamplona, Spain

[7]CREA - Council for Agricultural Research and Economics, Research Centre for Agriculture and Environment, 00198 Rome, Italy

[8]Swedish University of Agricultural Sciences, Department of Ecology, Box 7044, 75007 Uppsala, Sweden

[9]INRA Orléans, InfoSolUnit, Orléans,France

[10]Ecosys, INRA-AgroParisTech, Université Paris-Saclay, Campus AgroParisTech, 78850 Thiverval-Grignon, France

*Correspondence to:* Elisa Bruni (elisa.bruni@lsce.ipsl.fr)



**Abstract.** The 4 per 1000 initiative aims to promote better agricultural practices to maintain and increase soil organic carbon stocks for soil fertility, food security and climate change adaptation and mitigation. The most straightforward way to enhance soil organic carbon stocks is to increase carbon inputs to the soil.

In this study, we assessed the amount of organic carbon inputs that are necessary to reach a target of soil organic carbon stocks increase by 4‰ per year on average, for 30 years. We used the Century model to simulate soil organic carbon stocks in 14 European long-term agricultural experiments and assessed the required level of carbon inputs increase to reach the 4 per 1000 target. Initial simulated stocks were computed analytically assuming steady state. We compared modelled carbon inputs to different treatments of additional carbon used on the experimental sites (exogenous organic matter addition and one treatment with different crop rotations). We then analyzed how this would change under future scenarios of temperature increase. The model was calibrated to fit the control plot, i.e. conventional management without additional carbon inputs, and was able to reproduce the SOC stocks dynamics.

We found that, on average among the selected experimental sites, annual carbon inputs will have to increase by $43.15 \pm 5.05$ %, which is $0.66 \pm 0.23$ MgC ha$^{-1}$ per year (mean $\pm$ standard error), with respect to the control situation. The simulated amount of carbon inputs required to reach the 4‰ SOC increase was lower or similar to the amount of carbon inputs actually used in the majority of the additional carbon input treatments of the long-term experiments. However, Century might be overestimating the effect of additional C inputs on the variation of SOC stocks in some sites, since we found that treatments with additional carbon inputs were increasing by 0.25% on average among the experimental sites.

We showed that the modeled carbon inputs required to reach the target depended linearly on the initial SOC stocks. We estimated that annual carbon inputs would have to increase further due to temperature increase effect on decomposition rates, that is 54% for a 1°C warming and 120% for a 5°C warming.

## 1 Introduction

Increasing organic carbon (C) stocks in agricultural soils is beneficial for soil fertility and crop production and for climate change adaptation and mitigation. This consideration was at the basis of the 4 per 1000 (4p1000) initiative, proposed by the French Government during the 21$^{st}$ Conference of the Parties (COP21) on climate change. The 4p1000 initiative aims at promoting agricultural practices that enable the conservation of organic carbon in the soil (www.4p1000.org). Because soil organic carbon (SOC) stocks are two to three times higher than those in the atmosphere, even a small increase of the SOC pool can translate into significant changes in the atmospheric pool (Minasny et al., 2017). To demonstrate the importance of SOC, the initiative took as an example the fact that increasing global SOC stocks up to 0.4 m depth by 4p1000 (0.4%) per year of their initial value could offset the net annual $CO_2$ anthropogenic emissions to the atmosphere (Soussana, 2017). While increasing SOC stocks by 4p1000 annually is not a normative target of the initiative, this value can be taken as a reference to which current situations and alternative strategies are compared (e.g. Pellerin et al., 2017).



Strategies of conservation and expansion of existing SOC pools may be necessary but not sufficient to
mitigate climate change (Paustian et al., 2016). In this sense, increasing SOC stocks cannot be regarded as a
dispensation to continue business as usual, but rather as a wedge of negative greenhouse gases (GHG)
emissions (Wollenberg et al., 2016), as well as a strategy for improving most soils' resilience face to changes
in climate.
The potential to increase SOC stocks is particularly relevant in cropped soils, where the depletion of organic
matter with respect to the original non-cultivated situation has been assessed (Clivot et al., 2019; Goidts and
van Wesemael, 2007; Meersmans et al., 2011; Saffih-Hdadi and Mary, 2008; Sanderman et al., 2017; Zinn
et al., 2005) and where straightforward management practices can be implemented to promote the
conservation or increment of carbon in the soil (Chenu et al., 2019; Guenet et al., 2020; Paustian et al., 2016).
Moreover, increasing the organic carbon content in agricultural soils is known to improve their fertility and
water retention capacity (Lal 2008), indirectly enhancing agricultural productivity, food security and
eventually promoting a virtuous C cycle.
SOC stocks result from a balance between C inputs and C outputs. To increase SOC stocks one can either
increase C inputs to the soil (i.e. adding plant material or organic fertilizers) or reduce C outputs resulting
from mineralization and, in some cases, soil erosion. Increasing SOC stocks can be achieved via agricultural
practices such as retention of crop residues and organic amendments to the soil, cover cropping, diversified
rotations and agroforestry systems (Chenu et al., 2019). However, some of these practices only lead to local
*carbon storage* at field scale, rather than a net *carbon sequestration* from the atmosphere at larger scales. For
example, redistributing crop residues or organic fertilizers on a specific agricultural field rather than
spreading them over a larger landscape might induce local carbon storage increase, but does not remove
additional C from the atmosphere. In general, we can refer to carbon sequestration as the process of
transferring $CO_2$ from the atmosphere to the soil (Olson et al., 2014), while carbon storage more broadly
indicates the increase of SOC stocks over time and is not necessarily associated with net removal of GHG
from the atmosphere (Chenu et al., 2019).
Assessing the evolution of SOC stocks over time is important for estimating correctly the potential of SOC
storage in agricultural soils and evaluating management practices in terms of both SOC stocks increase and
sequestration potential. The dynamics of SOC stocks can be either measured in agricultural soils through
long-term experiments (LTEs) and soil monitoring networks or estimated via biogeochemical models
(Campbell and Paustian, 2015; Manzoni and Porporato, 2009). LTEs where SOC stocks and other
parameters, such as C inputs and climatic conditions, have been measured frequently are expensive and must
have been setup and kept on for a long time. For this reason, they are rare and unequally distributed across
the world. Extrapolating field data analysis from one region of the world to another can lead to wrong
estimations of the SOC storage potential in agricultural soils. In fact, distinct pedo-climatic conditions across
the world affect the potential SOC storage rate and capacity at different scales, as they imply different
mineralization kinetics and initial SOC contents (Chenu et al., 2019). Also, systems with low initial SOC
stocks like croplands may have a larger potential to re-store C than systems that have already high SOC





stocks (e.g. non-degraded grasslands), as noted by Minasny et al. (2017). Combining measurements of SOC
with models provides a wider applicability of the information collected in field trials. SOC model simulations
allow estimating the evolution of SOC stocks and their future trends to assess the potential gain of SOC at
global scale and following changes in agricultural practices. However, validity of models in the studied areas
has to be assessed and models need to be initialized (i.e. the initial size of SOC in the studied areas has to be
determined), often requiring the hypothesis that SOC is at equilibrium at the beginning of the experiment
(Luo et al., 2017; Xia et al., 2012).
Studying the feasibility and applicability of the 4p1000 initiative at site scale, means taking into account site-
specific conditions: historical land-use, pedo-climatic context and management practices. All these elements
will determine the additional organic matter inputs required to increase SOC stocks to a 4‰ annual rate.
Minasny et al. (2017) described opportunities and limitations of a 4‰ SOC increase in 20 regions across the
world. However, several authors (Baveye et al., 2018; van Groenigen et al., 2017; VandenBygaart, 2018)
argued that some of the examples described by Minasny et al. (2017) were not representative of wide-scale
agriculture and suggested that a 4‰ rate was not feasible in many practical situations (Poulton et al., 2018)
In this context, a few questions arise: how much should we increase C inputs to the soil to increase SOC
stocks by 4‰ per year? Is this amount attainable with currently implemented soil practices? And how is that
going to evolve in a future driven by climate change? In this study, we tried to answer these questions using
the biogeochemistry SOC model Century. We set the target of SOC stocks increase to 4‰ per year relatively
to the initial stocks, for 30 years of experiment. We simulated the SOC stocks in 14 different agricultural
LTEs around Europe and estimated the amount of additional carbon inputs required to reach the 4p1000
target. Finally, we evaluated the dependency of the required additional carbon inputs relatively to different
scenarios of increased temperature.
## 2    Materials and methods
### 2.1.  Experimental sites
We compiled data from 14 long-term experiments in arable cropping systems across Europe (Fig. 1), where
a total of 46 treatments increasing the inputs of C into the soil were performed and one control plot was
implemented (Table 1). The experiments lasted between 11 and 53 years (median value of 16 years) in the
period from 1956 to 2018. Most of the experiments had at least 3 replicates, except for the Italian site *Foggia*,
the French site *Champ Noël 3* and the British site *Broadbalk,* where no replicates were available. We selected
experiments with a duration of at least 10 years, where dry matter (DM) yields and soil organic carbon had
been measured at several dates. C inputs in all sites except from *Foggia* in Italy included exogenous organic
matter (EOM) addition, e.g. animal manure, household waste, sewage sludge or compost additions. In
*Foggia*, different rotations without organic matter addition were studied and compared to a wheat-only
treatment, considered as the control plot. The annual C inputs to the soil were substantially higher in the





rotations compared to the control. More information on crop rotations and carbon inputs for each treatment
can be found in Table 1.
Cropping systems found in the 60 treatments (14 control plots and 46 additional carbon inputs treatments)
were mainly cereal-dominated rotations (wheat, maize, barley and oat). In particular, four were cereal
monocultures (silage maize in *Champ Noël 3*, *Le Rheu 1* and *Le Rheu 2* and winter wheat in *Broadbalk*) and
four sites had rotations of different cereals (winter wheat and silage or grain maize in *Crécom 3 PRO*,
*Feucherolles*, *La Jaillière 2 PRO* and *Avrillé*). The other experiments rotated cereal crops with legumes
(chickpea, pea) and/or root crops (fodder beet, fodder rape and Swedish turnip), oilseed crops (sunflower and
oilseed rape), cover crops (mustard and rapeseed) and one rotation included tomatoes. Straw residues were
systematically exported except in French sites, where residues were sometimes incorporated into the soil as
accounted for in the carbon input calculations. All LTEs were under conventional tillage, which was
performed with a tractor, except in the case of *Ultuna* where it was performed manually. All experiments
were rainfed, except for *Foggia*, where tomatoes were irrigated in summer. The French experiments *Champ*
*Noël 3*, *Crécom 3 PRO*, *La Jaillière 2 PRO*, *Le Rheu 1* and *Trévarez* received optimal amounts of mineral
fertilizers both in the control plot and in the different organic matter treatments. All other experiments did
not receive any mineral fertilization. All control plots, a part from *Arazuri*, had decreasing SOC stock trends
(SOC approximated with a linear regression: $SOC = m \cdot t + SOC_0$, with average relative change: $\frac{m}{soc_0} \cdot$
$100 = -0.76$ %, $R^2 = 0.58$). Over the 46 treatments of additional carbon input, 19 exhibited increasing SOC
stocks at a higher ratio than 4‰ per year on average over the experiment length (Table 1). 13 treatments had
increasing SOC stocks, but at a lower ratio than 4p1000. The other 14 treatments with additional carbon
inputs had decreasing SOC stocks (MgC ha⁻¹). However, the decreasing trend was, in these cases, lower than
the decreasing trend in the respective control plot, on the majority of the treatments.
**Table 1: Summary of the agricultural experiments included in the study: crop rotations grown at site, amount of**
**carbon inputs (MgC ha⁻¹ per year) estimated from crop yields as in (Bolinder et al., 2007), type of treatments,**
**amount of additional organic carbon for each treatment (MgC ha⁻¹ per year) and mean annual SOC stocks**
**variation (%).**

| Site | ID Treatment | Rotations* | Carbon inputs from crop rotations | Treatment type | Additional carbon inputs | SOC annual variation |
|---|---|---|---|---|---|---|
| | | | MgC/ha/year | | MgC/ha/year | % |
| Champ Noël 3 | Min** | sM | 1.29 | Reference+N ** | 0 | -0.92 |
| (CHNO3) | LP | Silage maize | 1.49 | Pig manure | 0.79 | -0.89 |
| Colmar | T0 | wW/Mg/sB/S | 2.79 | Reference | 0 | -0.78 |
| (COL) | BIO1 | wW/Mg/sB/S | 3.93 | Biowaste | 1.01 | 0.15 |
| | BOUE1 | wW/Mg/sB/S | 3.96 | Sewage sludge | 0.49 | -0.61 |
| | CFB1 | wW/Mg/sB/S | 4.04 | Cow manure | 1.07 | -0.01 |





| | DVB1 | wW/Mg/sB/S | 4.00 | Green manure+Sewage sludge | 1.08 | 0.18 |
|---|---|---|---|---|---|---|
| | FB1 | wW/Mg/sB/S | 3.93 | Cow manure | 1.36 | -0.01 |
| Crécom 3 PRO (CREC3) | Min | wW/sM | 1.84 | Reference+N | 0 | -0.06 |
| | FB2 | wW/sM | 1.92 | Cow manure | 1.82 | 0.49 |
| | FV | wW/sM | 1.96 | Poultry manure | 0.47 | -1.46 |
| Feucherolles (FEU) | T0 | wW/ Mg | 2.22 | Reference | 0 | -0.66 |
| | BIO1 | wW/Mg | 3.44 | Biowaste | 2.21 | 3.60 |
| | DVB1 | wW/Mg | 3.45 | Green manure+Sewage sludge | 2.45 | 3.69 |
| | FB1 | wW/Mg | 3.55 | Cow manure | 2.28 | 1.36 |
| | OMR1 | wW/Mg | 3.45 | Household waste | 2.11 | 1.72 |
| Jeu-les-Bois (JEU) | M0 | wB/R/wW | 2.99 | Reference | 0 | -1.33 |
| | CFB1 | wB/R/wW | 2.89 | Cow manure | 1.1 | 1.61 |
| | CFB2 | wB/R/wW | 3.06 | Poultry manure | 1.94 | 1.52 |
| | FB2 | wB/R/wW | 3.11 | Cow manure | 2.43 | 0.99 |
| La Jaillière 2 PRO (LAJA2) | Min | sM/wW | 1.59 | Reference+N | 0 | -1.43 |
| | CFB | sM/wW | 1.25 | Cow manure | 1.14 | -0.88 |
| | CFP | sM/wW | 1.21 | Pig manure | 1 | -1.09 |
| | CFV | sM/wW | 1.31 | Poultry manure | 0.94 | -1.60 |
| | FB | sM/wW | 1.29 | Cow manure | 1.44 | -0.64 |
| | FP | sM/wW | 1.27 | Pig manure | 1.07 | -1.03 |
| | FV | sM/wW | 1.40 | Poultry manure | 0.93 | -1.59 |
| Le Rheu 1 (RHEU1) | Min | sM | 1.31 | Reference+N | 0 | -1.51 |
| | CFB1 | sM | 1.31 | Cow manure | 1.06 | -1.21 |
| Le Rheu 2 (RHEU2) | T0 | sM | 1.03 | Reference | 0 | -1.72 |
| | CFP1 | sM | 1.20 | Pig manure | 0.78 | -1.28 |
| | FP | sM | 1.30 | Pig manure | 1.62 | -0.74 |
| Arazuri (ARAZ) | DO_N0 | B/P/W/Sf/O | 0.98 | Reference | 0 | 1.00 |
| | D1_F1 | B/P/W/Sf/O | 1.40 | Sewage sludge | 2.82 | 0.40 |
| | D1_F2 | B/P/W/Sf/O | 1.41 | Sewage sludge | 1.4 | 1.22 |
| | D1_F3 | B/P/W/Sf/O | 1.44 | Sewage sludge | 0.78 | 1.22 |





| | | | | | | |
|---|---|---|---|---|---|---|
| | D2_F1 | B/P/W/Sf/O | 1.30 | Sewage sludge | 5.64 | 0.22 |
| | D2_F2 | B/P/W/Sf/O | 1.40 | Sewage sludge | 2.8 | 2.32 |
| | D2_F3 | B/P/W/Sf/O | 1.49 | Sewage sludge | 1.56 | 0.93 |
| Ultuna | P0_B | O/sT/Mu/sB/FB/OsR/W/FR/M | 1.03 | Reference | 0 | -0.52 |
| (ULTU) | S_F | O/sT/Mu/sB/FB/OsR/W/FR/M | 1.10 | Straw | 1.77 | -0.09 |
| | GM_H | O/sT/Mu/sB/FB/OsR/W/FR/M | 1.82 | Green manure | 1.76 | 0.11 |
| | PEAT_I | O/sT/Mu/sB/FB/OsR/W/FR/M | 1.14 | Peat | 1.97 | 2.17 |
| | FYM_J | O/sT/Mu/sB/FB/OsR/W/FR/M | 1.76 | Farmyard Manure | 1.91 | 0.69 |
| | SD_L | O/sT/Mu/sB/FB/OsR/W/FR/M | 0.82 | Sawdust | 1.84 | 0.56 |
| | SS_O | O/sT/Mu/sB/FB/OsR/W/FR/M | 2.59 | Sewage sludge | 1.84 | 1.36 |
| Broadbalk | 3_Nill | wW | 0.36 | Reference | 0 | -0.09 |
| (BROAD) | 19_Cast | wW | 0.65 | Castor meal | 0.43 | 0.42 |
| | 22_FYM | wW | 2.07 | Farmyard Manure | 3 | 0.38 |
| Foggia | T0 | W | 1.56 | Reference | 0 | -0.86 |
| | Dw-Dw-Fall | W/W/F | 2.13 | Rotation | 0.57 | 0.01 |
| | Dw-Fall | W/F | 1.95 | Rotation | 0.39 | -0.33 |
| | Dw-Oa-Fall | W/O/F | 2.20 | Rotation | 0.64 | -0.33 |
| | Dw-Dw-Cp | W/W/C | 2.53 | Rotation | 0.97 | -0.15 |
| | Dw-Dw-To | W/W/T | 2.57 | Rotation | 1.01 | -0.59 |
| Trévarez | Min | RG/Mg/wW/sM | 1.94 | Reference+N | 0 | -0.66 |
| (TREV) | FB | RG/Mg/wW/sM | 2.04 | Cow manure | 1.52 | -0.39 |
| | FP | RG/Mg/wW/sM | 2.02 | Pig manure | 1.18 | -0.18 |
| Avrillé | T12TR | wW/sM | 2.25 | Reference | 0 | -1.18 |
| (AVRI) | T2TR | wW/sM | 2.36 | Cow manure | 1.68 | -0.76 |

*Crops: sM = silage Maize, Mg= Maize grain, wW = winter Wheat, W = Wheat,

sB = spring Barley, wB = winter Barley, B = barley, S = sugarbeet,

R = Rapeseed, Sf = Sunflower, O = Oats, P = Pea, sT = Swedish Turlip, Mu =

Mustard, DF = Fodder Beet, OsR = Oilseed Rape, FR = fodder Rape,

F = green Fallow, C = Chickpeas, T = Tomato, RG = Ray Grass

**Optimal amounts of mineral fertilizers added to the control

plot and to all other treatments in the experiment




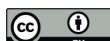

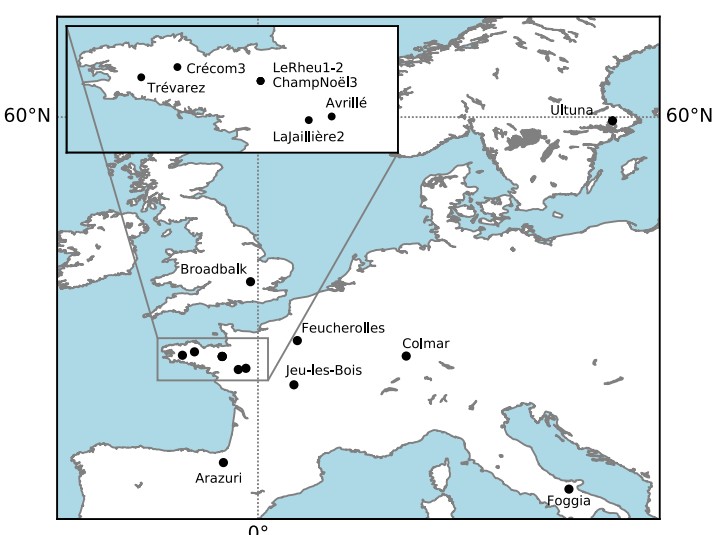


**Figure 1: Location of the 60 field trials distributed among the 14 cropland experiments around Europe.**

**2.1.1. Climate forcing**
Mean temperature of the sites ranged from a minimum of 5.7 °C to a maximum of 15.5 °C, while mean soil
humidity to approximately 20 cm depth was 21.9 $kg_{H2O}$ $m^{-2}_{soil}$ for the whole dataset (Table 2). When available,
observed daily air temperature was used as an approximation of soil temperature. Otherwise, land-
atmosphere model ORCHIDEE was used to simulate soil surface temperature and soil humidity at site-scale
(Krinner et al., 2005). ORCHIDEE simulations were run over each site using a 3-hourly global climate dataset
at 0.5° (GSWP3 http://hydro.iis.u-tokyo.ac.jp/GSWP3/). Plant cover was set to C3 plant functional type
(PFT) for agriculture.
**Table 2: Mean annual values of temperature (C°) and soil humidity to approximately 20 cm depth ($kg_{H2O}$ $m^{-2}$)**
**simulated with ORCHIDEE model over each experimental site. Measured pH, bulk density (g $cm^{-3}$), clay (%) and**
**initial SOC stocks in the control plots (MgC $ha^{-1}$) on the agricultural fields. Reference papers for each site are**
**indicated. [1]For *Arazuri*, data were directly provided by the Spanish Mancomunidad de la Comarca de Pamplona.**

| Sites | Reference paper | Coordinates | Years | Mean annual Temperature | Mean annual soil humidity | pH | Bulk density | Clay | Initial SOC stocks |
|---|---|---|---|---|---|---|---|---|---|
| | | | | °C | kg H₂O m² | | g cm⁻³ | % | MgC ha⁻¹ |
| Champ Noël 3 | (Clivot et al., 2019) | 48.09° N, 1.78° W | 1990 - 2008 | 12.1 | 21.6 | 6.3 | 1.35 | 15.1 | 40.57 |
| Colmar | (Clivot et al., 2019) | 48.11° N, 7.38° E | 2000 - 2013 | 9.6 | 24.6 | 8.33 | 1.3 | 23.1 | 54.33 |





| | | | | | | | | | |
|---|---|---|---|---|---|---|---|---|---|
| Crécom 3 PRO | (Clivot et al., 2019) | 48.32 ° N, 3.16 ° W | 1986 - 2008 | 11.8 | 22.9 | 6.15 | 1.36 | 14.6 | 62 |
| Feucherolles | (Clivot et al., 2019) | 48.88° N, 1.96° E | 1998 - 2013 | 11.9 | 21.2 | 6.73 | 1.32 | 15.6 | 39.78 |
| Jeu-les-Bois | (Clivot et al., 2019) | 46.68° N, 1.79° E | 1998 - 2008 | 12.2 | 22.1 | 6.27 | 1.52 | 10 | 48.53 |
| La Jaillière 2 PRO | (Clivot et al., 2019) | 47.44° N, 0.98° W | 1995 - 2009 | 12.7 | 20.5 | 6.8 | 1.37 | 20.8 | 32.42 |
| Le Rheu 1 | (Clivot et al., 2019) | 48.09° N, 1.78° W | 1994 - 2009 | 12.2 | 21.8 | 5.85 | 1.27 | 16.4 | 36.23 |
| Le Rheu 2 | (Clivot et al., 2019) | 48.09 N, 1.78 W | 1994 - 2009 | 12.2 | 21.8 | 6.05 | 1.28 | 13.9 | 36.53 |
| Arazuri[1] | - | 42.81° N, 1.72° W | 1993 - 2018 | 12.7 | 20.4 | 8.6 | 1.67 | 27.9 | 55.39 |
| Ultuna | (Kätterer et al., 2011) | 59.82° N, 17.65° E | 1956 - 2008 | 5.7 | 22.6 | 6.23 | 1.4 | 36.5 | 41.72 |
| Broadbalk | (Powlson et al. 2012) | 51.81° N, 0.37° W | 1968 - 2015 | 10.2 | 21.5 | 7.8 | 1.25 | 25 | 24.84 |
| Foggia | (Farina et al., 2017) | 41.49° N, 15.48° E | 1992 - 2008 | 15.5 | 22.4 | 8.1 | 1.32 | 41 | 63.22 |
| Trévarez | (Clivot et al., 2019) | 48.15° N, 3.76° W | 1986 - 2008 | 11.8 | 23.4 | 6.01 | 1.48 | 19.2 | 115.33 |
| Avrillé | (Clivot et al., 2019) | 47.50° N, 0.60° W | 1983 - 1991 | 12.0 | 20.2 | 6.59 | 1.4 | 17.6 | 54.46 |

**2.1.2. Soil characteristics**
The sampling depth of the experiments varied between 20 and 30 cm. SOC stocks were measured in 3 – 4
replicates, apart from *Foggia* and *Champ Noël 3* experiments, were no replicates were available. In
*Broadbalk* experiment, SOC was measured in each plot using a semi-cylindrical auger where 10-20 cores
were taken from across the plot and bulked together (more details can be found on the e-RA website[1]). The
clay content ranged from 10% (*Jeu-les-Bois*) to 41% (*Foggia*). Soil pH varied from a minimum of 5.85 in *Le*
*Rheu 1* to a maximum of 8.33 in *Colmar*. The average bulk density (BD) in the control plots was 1.38 g cm
$^{-3}$. SOC stocks (MgC ha$^{-1}$) were calculated at each site using the following equation:
$SOC\ (MgC\ ha^{-1}) = SOC(\%) \cdot BD(g\ cm^{-3}) \cdot sampling\ depth\ (cm),$          (1)
where SOC (%) is the concentration of organic carbon in the soil, BD is the average bulk density of the
experimental plot. It should be noted that the application of EOMs might induce differences in bulk density
with time, which in turn affects the calculations of SOC stocks. No adjustment was made in this sense, since

---

[1] www.era.rothamsted.ac.uk





data on the evolution of BD was available only for a few sites. This might explain differences between the
SOC stocks calculated for *Broadbalk* in this paper and those found by Powlson et al. (2012) in the same site,
by adjusting soil weights to observed decreases in top soil BD due to accumulating farmyard manure (FYM).
Initial SOC stocks values in the control plot and mean climate variables for each site are reported in Table 2.
**2.2. Century model**
**2.2.1. Model description**
Soil carbon dynamics in a soil organic matter model with first-order kinetics can be mathematically described
by the following first-order differential matrix equation:
$\frac{\mathrm{d}\boldsymbol{SOC}(\mathrm{t})}{\mathrm{d}t} = \boldsymbol{I} + \mathbf{A} \cdot \boldsymbol{\xi}_{\mathbf{TWLCI}}(\mathrm{t}) \cdot \mathbf{K} \cdot \boldsymbol{SOC}(\mathrm{t}),$          (2)
where $\boldsymbol{I}$ is the vector of the external carbon inputs to the soil system, with four nonzero elements (Fig. 2).
The second term $\mathbf{A} \cdot \boldsymbol{\xi}_{\mathbf{TWLCI}}(\mathrm{t}) \cdot \mathbf{K} \cdot \boldsymbol{SOC}(\mathrm{t})$ of the equation represents organic matter decomposition rates
(diagonal matrix $\mathbf{K}$), losses through respiration ($\boldsymbol{\xi}_{\mathbf{TWLCI}}(\mathrm{t})$) and transfers of C among different SOC pools
($\mathbf{A}$) (see Appendix A). We used the daily time-step version of the soil organic matter (SOM) model Century
(Parton et al., 1988) to simulate the amount of carbon inputs required to reach a 4‰ annual increase of soil
organic carbon storage over 30 years. The Century model has been successfully applied to long-term
experiments and has been validated for different ecosystem types (Bortolon et al., 2011; Cong et al., 2014;
Parton et al., 1993). The original version of Century simulates the fluxes of SOC depending on soil relative
humidity, temperature and texture (as a percentage of clay). As shown in Fig. 2, the model is discretized into
7 compartments that exchange carbon with each other: 4 pools of litter (aboveground metabolic, belowground
metabolic, aboveground structural and belowground structural) and 3 pools of soil organic carbon (active,
slow and passive). The litter carbon is partially released to the atmosphere as respired $CO_2$ and partially
converted to soil organic matter in the active, slow and passive pools (see Table S1 in the supporting
information for default Century parameters). The decomposition rate of C in the $i^{th}$ pool depends on climatic
conditions, litter and soil characteristics and is calculated using environmental response functions, as follows:
$\xi_{TWLCl}(t)_i \cdot K_i = k_i \cdot f_T(t) \cdot f_W(t) \cdot f_{L\,i} \cdot f_{Clay\,i},$          (3)
where $i = 1, \dots, 7$ is one of the aboveground (AG) and belowground (BG) metabolic and structural litter
pools, and the active, slow and passive SOC pools; $K_i$ is the $(K)_{ii}$ element of the diagonal matrix $\mathbf{K}$ in Eq.
(2); $k_i$ is the specific mineralization rate of pool $i$, $f_T(t)$ is a function of daily soil temperature, $f_W(t)$ is a
function used as a proxy to describe the effects of soil moisture, $f_{L\,i}$ is a reduction rate parameter acting on
the AG and BG structural pools only, depending on the lignin concentration in the litter and $f_{Clay\,i}$ is a
reduction rate function of clay on SOC mineralization in the active pool. The temperature function $f_T(t)$
describes the exponential dependence of soil decomposition on surface temperature, through the $Q_{10}$
relationship that was first presented by M. J. H. van't Hoff in 1884:
$f_T(t) = Q_{10}^{\frac{(T(t) - T_{ref})}{10}},$          (4)



where $Q_{10}$ is the temperature coefficient, usually set to 2 and $T_{ref}$ is the reference temperature of 30 ˚C. The
$Q_{10}$ factor is a measure of the soil respiration change rate as a consequence of increasing temperature by 10˚.
The other environmental response functions are described in Appendix A.

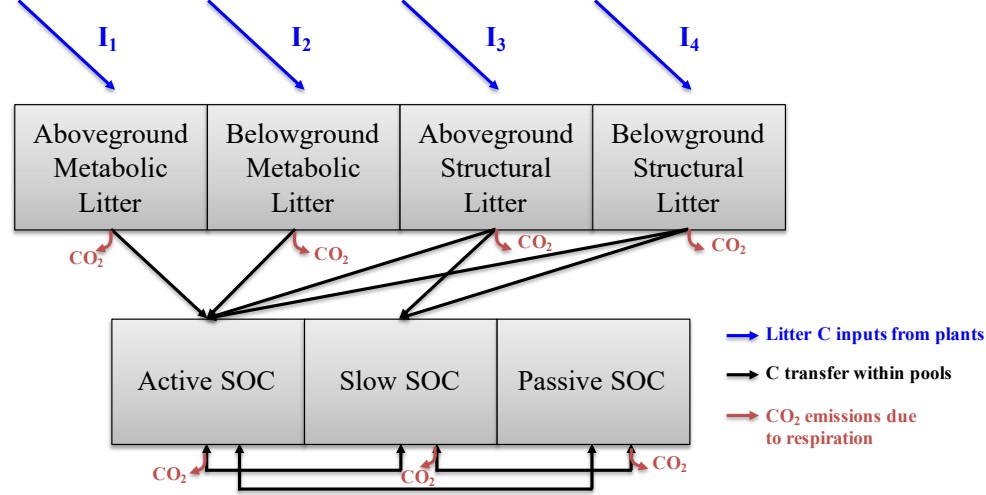


**Figure 2: Representation of litter and soil organic carbon (SOC) pools in Century. The model takes as inputs litter carbon from plants (aboveground metabolic ($I_1$), belowground metabolic ($I_2$), aboveground structural ($I_3$) and belowground structural ($I_4$)). A certain fraction of carbon can be transferred from one pool to another and each time a transfer occurs, part of this carbon is respired and leaves the system to the atmosphere as $CO_2$. The SOC active pool receives carbon from each litter pool, while only the structural material is transferred to the SOC slow pool. Litter material never goes directly to the SOC passive pool while the three SOC pools exchange C within each other.**

234       **2.2.2. Model initialization**

The initialization of the model consists in specifying the initial sizes of the SOC pools. Here, we assumed
initial pools are in equilibrium with carbon inputs before the experiments, in absence of knowledge about
past land use and climate making initial pools different from steady state (Sanderman et al., 2017). Then,
initialization can be done either by running the model iteratively for thousands of years to approximate the
steady state solution (numerical spin-up), or semi-analytically by solving the set of differential equations that
describe the carbon transfers within model compartments (Xia et al., 2012). We solved the matrix equation
by inverse calculations for determining pools sizes at steady state, as in Xia et al. (2012) and Huang et al.
(2018). These authors demonstrated that the matrix inversion approach exactly reproduces the steady state
and SOC dynamics of the model. By enhancing the computational performance of the simulations, this
technique enables the analysis of system properties and facilitates studying model behavior. It allowed us to
perform the optimization of model parameters, the sensitivity analysis of SOC to climatic variables and the
quantification of model outputs uncertainties through Monte-Carlo (MC) iterative procedures. We solved the
matrix equation by using its semi-analytical solution and the following algorithm: 1) calculating annual
averages of matrix items obtained by Century simulations, driven by 30 years of climatic forcing; 2) setting



Eq. (2) to zero to solve the state vector **SOC**. For each agricultural site, the 30 years of climate forcing were
set as the 30 years preceding the beginning of the experiment, and the litter input estimated from observed
vegetation was set to be the average litter input in the control plot over the experiment duration.
### 2.2.3. Carbon inputs
The allocation of C in the different litter pools was estimated with the approach firstly described by Bolinder
et al. (2007) for Canadian experiments and then adapted by Clivot et al. (2019) to the same French sites we
use in this study. This methodology allows splitting C inputs from crop residues after harvest into
aboveground and belowground C inputs, using measured dry matter yields and estimations of the shoot-to-
root ratio (S:R) and harvest indexes (HI) of the crops (see Fig. 3). The aboveground plant material is estimated
as the harvested part of the plant ($C_P$), which is exported from the soil, plus the straw and stubble that are left
in the soil after harvest ($C_S$). The harvested part consists of the measurements of dry matter yields ($Y_P$), while
the straw and stubble are estimated using the HI coefficient of the different crops in the rotation (Bolinder et
al., 2007). We assumed that the values used in Clivot et al. (2019) for the HI compiled from French
experimental sites were applicable to all the sites in our dataset, which mainly include temperate sites over
Europe. When these values were not available for some crops, they have been directly derived from Bolinder
et al. (2007) or other sources in the literature (S:R ratio for fallow from Mekonnen, Buresh, and Jama (1997)
and tomato from Lovelli et al. (2012)). When straw was exported from the field, we considered that only a
fraction of $C_S$ was left on the soil. This fraction was set to 0.4 for all sites and to 0.2 in *Ultuna*, where almost
no stubble was left on the soil, since plots were harvested by hand and crops were cut at the soil surface. We
considered a carbon content of 0.44 gC gDM$^{-1}$ in the aboveground plant material (Redin et al., 2014) and 0.4
gC gDM$^{-1}$ in the belowground part material (Bolinder et al., 2007). We used the asymptotic equation of Gale
and Grigal (1987) to determine the cumulative BG input fraction from the soil surface to a considered depth:
$$BG_{F\,depth} = 1 - \beta^{depth}, \tag{5}$$
where $\beta$ is a crop-specific parameter determined using the root distributions for temperate agricultural crops,
reported in Fan et al. (2016) and Clivot et al. (2019). The depth was set to 30 cm, since it was the depth at
which soil samples were taken in the majority of the sites. For more details on the carbon inputs allocation
method and the allometric functions involved, see Bolinder et al. (2007) and Clivot et al. (2019).



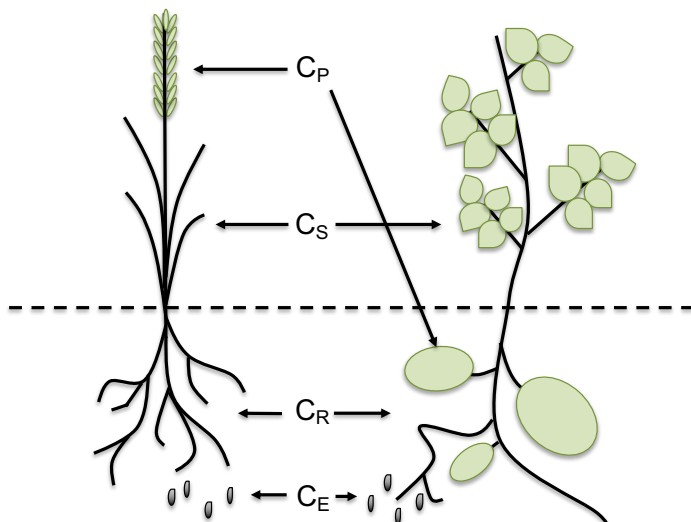


**Figure 3: Adapted from (Bolinder et al., 2007). Representation of the distribution of carbon in the different parts
of the plant: $C_P$ represents the carbon in the harvested product (grain, forage, tuber); $C_S$ is the carbon in the
aboveground residues (straw, stover, chaff); $C_R$ is the carbon present in roots and $C_E$ represents all the extra-root
carbon (including all root-derived materials not usually recovered in the root fraction).**

### 2.2.4. Model calibration: optimization of the metabolic:structural fractions of the litter inputs

In the Century model, AG and BG carbon inputs need to be further separated into metabolic and structural

fractions, according to the lignin to nitrogen (L:N) ratio. Because the L:N ratio was not available for all the

crops in the database, we fitted model simulations to observed SOC dynamics for the control plot of each

site, i.e. the reference plot without additional carbon inputs, in order to get the metabolic:structural (M:S)

fraction of the AG and BG carbon inputs. We used the sequential least-squares quadratic programming

function in Python (SciPy v1.5.1, scipy.optimize package with method='SLSQP'), a nonlinear constrained,

gradient-based optimization algorithm (Fu et al., 2019). We successfully performed the optimization on 13

sites, where at least three measures of SOC stocks were available. For *Jeu-les-Bois*, which includes two SOC

measurements only, we decided to use the same optimized values as for *Feucherolles*, which has similar

pedoclimatic conditions and crop rotations. The optimization consisted in minimizing the following function:

$$J_{fit} = \sum_{i=1}^{n} \frac{\left(SOC_i^{model} - SOC_i^{obs}\right)^2}{\sigma^2{}_i^{SOC_{obs}}}, \qquad (6)$$

where $i=1,...,n$ is the year of the experiment, $SOC_i^{model}$ (MgC ha⁻¹) is the SOC simulated with Century for

year $i$, $SOC_i^{obs}$ (MgC ha⁻¹) is the observed SOC for year $i$ in the control plot and $\sigma^2{}_i^{SOC_{obs}}$ is the variance of

the $SOC_i^{obs}$ estimated from the different replicates. When replicates were not available, we recalculated

$\sigma^{2 SOC_{obs}}$ as the variance amongst $SOC^{obs}$ samples of the whole experiment. The optimized M:S values are





reported in Table 3  and represent the average quality of litter carbon in the rotating crops along the duration
of the experiments that match control SOC data at each site.

**Table 3: Optimized values of the aboveground metabolic (AM), aboveground structural (AS), belowground metabolic (BM) and belowground structural (BS) fractions of the litter inputs and the Q10 and reference temperature (˚C) parameters.**

| Site | AM | AS | BM | BS | $Q_{10}$ | Reference temperature |
|------|----|----|----|----|------|----------------------|
|      |    |    |    |    |      | ˚C |
| CHNO3 | 0.85 | 0.15 | 0.26 | 0.74 | 5.0 | 21.2 |
| COL | 0.85 | 0.15 | 0.57 | 0.43 | 2.0 | 30.0 |
| CREC3 | 0.15 | 0.85 | 0.29 | 0.71 | 2.0 | 30.0 |
| FEU | 0.85 | 0.15 | 0.52 | 0.48 | 5.0 | 21.6 |
| JEU* | 0.85 | 0.15 | 0.52 | 0.48 | 5.0 | 21.6 |
| LAJA2 | 0.85 | 0.15 | 0.72 | 0.28 | 5.0 | 21.5 |
| RHEU1 | 0.85 | 0.15 | 0.49 | 0.51 | 5.0 | 21.3 |
| RHEU2 | 0.85 | 0.15 | 0.32 | 0.68 | 5.0 | 21.3 |
| ARAZ | 0.53 | 0.47 | 0.53 | 0.47 | 3.0 | 30.0 |
| ULTU | 0.85 | 0.15 | 0.85 | 0.15 | 2.2 | 30.0 |
| BROAD | 0.42 | 0.58 | 0.15 | 0.85 | 2.9 | 30.0 |
| FOGGIA | 0.15 | 0.85 | 0.15 | 0.85 | 5.0 | 27.1 |
| TREV1 | 0.15 | 0.85 | 0.15 | 0.85 | 5.0 | 23.0 |
| AVRI | 0.85 | 0.15 | 0.76 | 0.24 | 2.0 | 30.0 |

**2.2.5. Model calibration: optimization of temperature dependency parameters**
We optimized the $Q_{10}$ and daily soil reference temperature parameters, which affect SOC decomposition.
The $Q_{10}$ factor is fixed to 2 in Century. However, many authors have shown that $Q_{10}$ measurements vary with
pedoclimatic conditions and vegetation activity (Craine et al., 2010; Lefèvre et al., 2014; Meyer et al., 2018;
Wang et al., 2010). For this reason and to correctly reproduce interregional variations among the sites in the
dataset, we decided to optimize both the $Q_{10}$ and reference temperature parameters to better fit the SOC
dynamics (MgC ha$^{-1}$) of each agricultural site at control plot. We decided to bind the $Q_{10}$ between 1 and 5,
following the variation of $Q_{10}$ found by Wang et al. (2010) over 384 samples collected in the Northern
Hemisphere. The reference temperature ranged between 10 and 30˚C. We used the SLSQP optimization
algorithm and the cost function of Eq. (6) to perform the optimization, which was successful in 13 sites and
we assigned the values obtained from the optimization of *Feucherolles* to *Jeu-les-Bois*, where SOC
measurements were too sparse to perform a two-dimensional optimization. Optimized values of $Q_{10}$ and
reference temperature are reported in Table 3 .
Model performance in the control plot was evaluated using two residual-based metrics. The first one is the
Mean Squared Deviation (MSD), decomposed into its three components to help locating the source of error





of model simulations: the Squared Bias (SB), the Non-Unity slope (NU) and the Lack of Correlation (LC).

The second metrics used is the Normalized Root Mean Squared Deviation (NRMSD) (see Appendix B).

### 2.3. 4p1000 analysis

#### 2.3.1. Optimization of C inputs to reach the 4p1000 target

After the spin-up to steady state, the model was set to calculate the SOC stocks dynamics of the control plot and the carbon inputs for virtual treatments, assuming an average increase of SOC stocks by 4‰ per year over 30 years. 30 years is considered as a period of time over which the variation of SOC can be detected correctly. During this period length, we supposed the soil was fed with constant amounts of carbon inputs from plant material. For the control, we derived carbon inputs from measurements of DM yields and calculated the annual mean over the whole experiment length. For the virtual treatments, we used an optimization algorithm to calculate the required amount of carbon inputs to reach a linear increase of SOC storage by 4‰ per year above the SOC stock at the start of the simulation. Mathematically, we minimized the following function:

$$J_{4p1000} = \left| SOC_0 \cdot (1 + 0.004 \cdot 30) - SOC_{30}^{model}(I) \right|, \tag{7}$$

where $I$ is the 1x4 vector of C inputs to minimize over, $SOC_0$ is the initial soil organic carbon stock and $SOC_{30}^{model}(I)$ is the soil organic carbon stock after 30 years of simulation. During the optimization, the metabolic:structural fractions were allowed to vary to estimate the quality of the optimal carbon inputs. Instead, we kept the aboveground:belowground ratio of the C inputs fixed to its initial value, to bind the model in order to represent agronomically plausible C inputs. In fact, if not bound, the model tends to increase the belowground C fraction to unrealistic values (assuming the same crop rotations persisted on site). On the other hand, keeping the aboveground:belowground ratio fixed implies that the simulated additional C inputs will be spread equally on surface and belowground. As for the previous optimizations, we used the Python function SLSQP to solve the minimization problem. The outcome of the optimization is a 4x1 vector ($I_{opt}$) representing the amount of carbon in the four litter input pools that matches the 4p1000 rate target.

#### 2.3.2. Uncertainties quantification

Uncertainties of model outcomes were quantified using a Monte-Carlo approach. We initially calculated the standard error (SE) of the mean C inputs derived from yield measurements for each experimental site:

$$SE = \sqrt{\frac{\sigma^2_I}{s}}, \tag{8}$$

where $\sigma^2_I$ is the variance of the estimated C input from yield measurements and s is the size of the experiment. If not available, we calculated $\sigma^2_I$ as the average relative variance of C inputs among the control plots. We therefore randomly generated N vectors of C inputs ($I$) around the calculated standard error and performed the 4p1000 optimization N times, each time using one of the generated vectors I as a prior for the





optimization. To correctly assess the uncertainty over the required carbon inputs we set N to 50 (Anderson,
1976). The standard error of model outputs was calculated with Eq. (8), where the variance was set as the
variance of the modelled carbon outputs and the experiment size (s) to 50.

### 2.3.3. Sensitivity analysis to temperature

We considered two representative concentration pathways (RCPs) of global average surface temperature
change projections (IPCC, 2015). The first scenario (RCP2.6) is the one that contemplates stringent
mitigation policies and predicts that average global land temperature will increase by 1˚C during the period
2081-2100, compared to the mean temperature of 1986-2005. The second scenario (RCP8.5) estimates an
average temperature increase of +4.8˚C, compared to the same period of time. We ran two simulations of
increasing temperature scenarios with Century. We considered the same initial conditions as the standard
simulations, hence running the spin-up with the average soil temperature and relative humidity of the 30
years preceding the experiments. Then, we increased daily temperature by 1˚C (AS1) and 5˚C (AS5) for the
entire simulation length, to assess the variation of the required carbon inputs to reach the 4p1000 target,
mimicking RCP2.6 and RCP8.5 scenarios respectively.

## 3    Results

### 3.1.    Fit of calibrated model to control SOC values

Modelled and measured SOC stocks in the control plot were compared to evaluate the capability of the
calibrated version of Century to reproduce the dynamics of carbon stocks in the selected sites. As shown in
Fig. 4.b, the normalized root mean square error of the control plot SOC stocks is lower than 15% for all the
treatments, indicating that overall model simulations fitted quite well the observed SOC stocks (observed
SOC stocks variance was 16.3% on average in the control plots). Fig 4.a, provides the values of the three
components of the MSD indicator for each site. It can be noticed that the LC and NU components are the
highest contributors to MSD. This means that the major sources of error are the representation of the data
shape and magnitude of fluctuation among the measurements. The highest NRMSD can be found in *Le Rheu*
*1* and *Le Rheu 2* (around 12% and 14% respectively). In these sites the model seems to better capture the
shape of the data (low LC compared to the other sites), but it misses the representation of mean C stock (high
SB) and data scattering (high NU) of the experimental profiles.



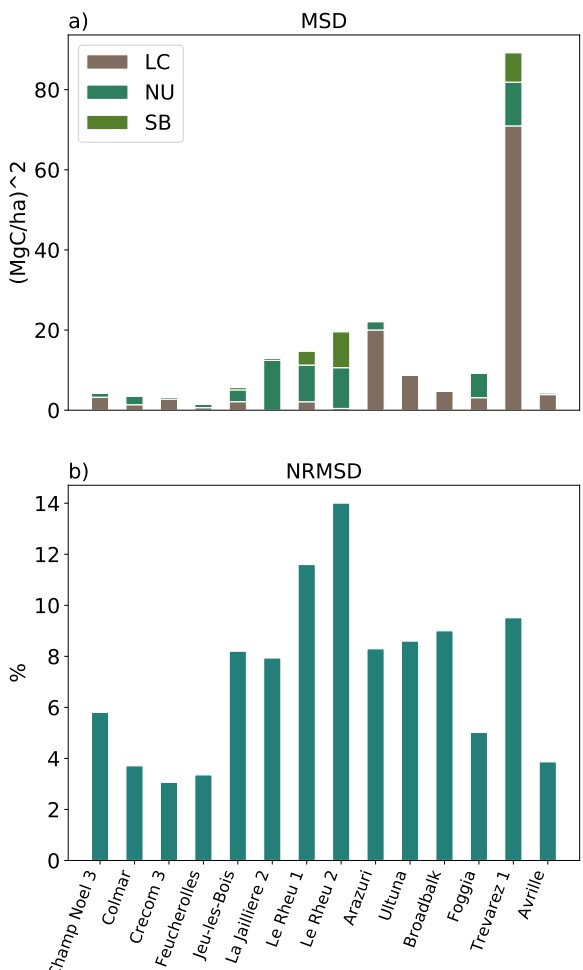

**Figure 4: a) Decomposed mean squared deviation (MgC ha⁻¹)² in control plots for all sites. LC = Lack of Correlation, NU = Non-Unity slope and SB = Squared Bias. b) Normalized root squared deviation (%) in control plots for all sites.**

### 3.2. Estimates of additional carbon inputs and SOC changes

#### 3.2.1. Virtual C inputs to reach the 4p1000

Figure 5 represents the average percentage change of carbon inputs optimized to reach the 4‰ annual increase of SOC stocks, among the whole sites. The increase of carbon inputs is given for each litter pool. On average, a $43.15 \pm 5.05$ % (mean $\pm$ SE across sites) increase of total annual carbon inputs compared to the current situation in the control plot, is required to meet the 4p1000 target. In terms of absolute values, this represents an additional $0.66 \pm 0.23$ MgC ha⁻¹ inputs per year, i.e. $2.35 \pm 0.21$ MgC ha⁻¹ total inputs per





year (equivalent approximately to 4.05 $\pm$ 0.36 MgDM ha$^{-1}$ per year). What stands out in the graph, is that
globally the aboveground structural litter pool should be more than doubled, while the other pools need only
to increase by about half of their initial value. In terms of absolute values, the structural aboveground biomass
(which was initially 0.29 MgC ha$^{-1}$ per year on average in the control treatments) would need an additional
0.18 MgC ha$^{-1}$ per year to reach the 4p1000; the metabolic aboveground (initially 0.70 MgC ha$^{-1}$ per year on
average) needs an additional 0.14 MgC ha$^{-1}$ per year; structural and metabolic belowground biomass (initially
0.65 and 0.52 MgC ha$^{-1}$ per year) require an additional C input corresponding to 0.21 and 0.13 MgC ha$^{-1}$ per
year respectively.
Analysis of the SOC pools evolution in the runs with optimized inputs to match the 4p1000 increase rate,
indicates that the active and slow pools increased by 0.58% and 0.61% per year respectively, while the
passive pool increased annually by 0.01% (Fig. 6). In absolute values, the slow compartment contributed the
most to the increase of SOC during the 30 years runs, as it increased by 2.7 MgC ha$^{-1}$ on average among the
sites. This corresponds to a storage efficiency for the 30 years of simulation of approximately 13.7 % in the
slow pool, compared to a storage efficiency of 0.5% and 0.34% in the active and in the passive pools
respectively.
We found a high linear relation (R$^2$=0.80) between observed initial SOC stocks and optimized carbon inputs
(Fig. 7). It is logical and expected that for low initial SOC stocks in steady state, a small increase of carbon
inputs is sufficient to reach the 4p1000 target. Conversely, when SOC is high at the beginning of the
experiment (e.g. *Trévarez*) much higher C inputs must be employed since our target increase rate is a relative
target. The regression line that emerges from the cross sites' relationship can be written as:
$I^{4p1000} = 0.013 \cdot SOC_0^{obs} + 0.001,$                                                              (9)
where $I^{4p1000}$ are the simulated C inputs needed to reach the 4p1000 target ($MgC\ ha^{-1}$ per year) and
$SOC_0^{obs}$ ($MgC\ ha^{-1}$) is the observed initial SOC stock. This result means that site differences in $Q_{10}$ and
decomposition rates are less influential than initial SOC in determining the optimal input increase to reach
the 4‰ per year target.





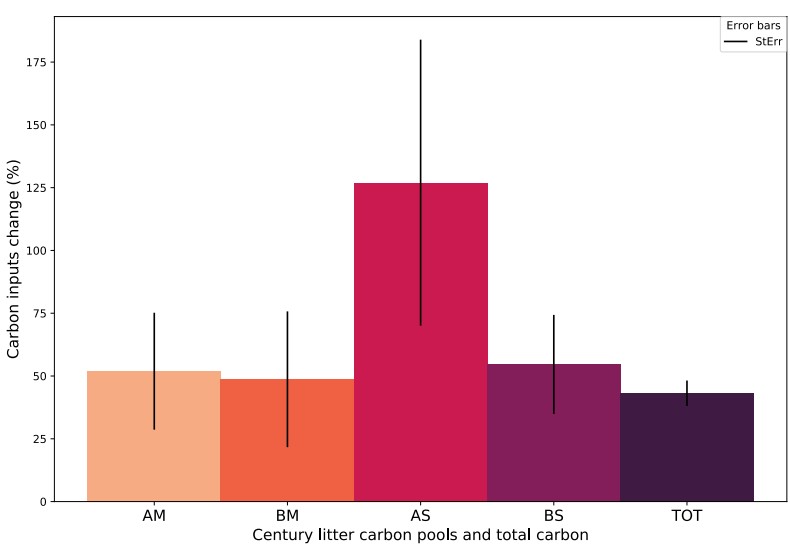


**Figure 5: Sites average percentage change of carbon inputs needed to reach the 4p1000 (TOT), separated into the four litter input pools. AM = aboveground metabolic, BM = belowground metabolic, AS = aboveground structural, BS = belowground structural and TOT = total litter inputs. Error bars indicate the standard error.**

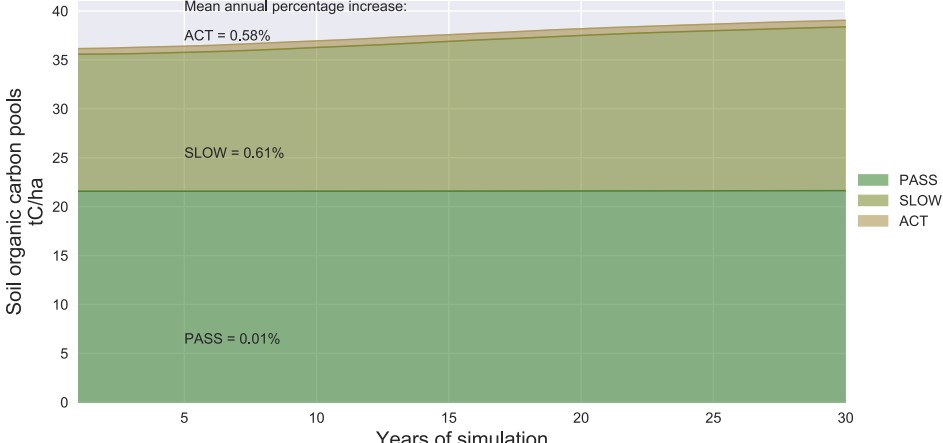

**Figure 6: Sites average soil organic carbon pools (ACT = active, SLOW = slow and PASS= passive) evolution**
**(MgC ha$^{-1}$) over the 30 years of simulation to reach the 4p1000 target. In the graph the mean percentage increase**
**is given for each SOC pool.**


EGU Open Access

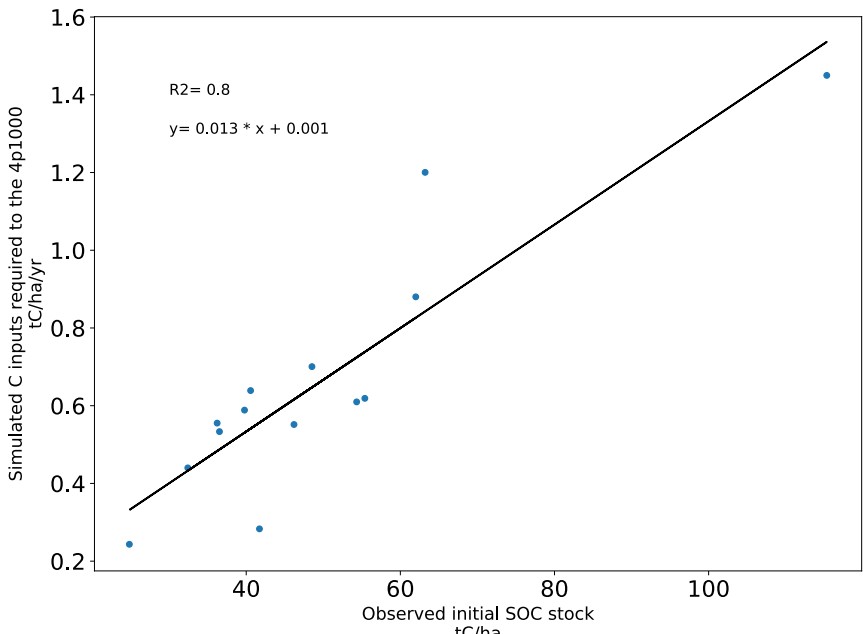


**Figure 7: Correlation between initial observed SOC stocks (MgC ha⁻¹) and modelled carbon inputs needed to reach the 4p1000 target (MgC ha⁻¹ year⁻¹). The correlation coefficient (R²) is 0.80 and the regression line is y = 0.013·x+0.001.**

**3.2.2. Virtual versus actual C inputs in the experimental carbon treatments**

In Fig. 8 we compare the virtual inputs required to reach the 4p1000 target to the actual inputs used across the 46 treatments of additional carbon. The additional carbon (MgC ha⁻¹ per year) shown in the graph for all experimental treatments refers to exogenous organic amendments, plus additional carbon due to increased crop yields, relatively to the reference plot. The most striking result emerging from the data is that modelled additional C inputs are systematically lower or similar to at least one treatment of additional C in all sites, except for *Foggia*. In *Foggia* experiment, different crop rotations were compared and no additional exogenous organic matter was incorporated to the soil. Here, none of the rotations had sufficient additional C content (compared to the control wheat-only treatment), to meet the required OC input level predicted by Century for a 4p1000 increase rate. Overall, 86.91% of the experimental treatments used higher amounts of carbon inputs compared to the modelled need of additional carbon inputs in the same site. For the other treatments, the difference between simulated and observed additional C input was not significant. On average, in the experimental treatments were applied 1.52 MgC ha⁻¹ per year and SOC stocks were found to be increasing by 0.25% per year. Modelled additional carbon input to reach the 4p1000 was 0.66 MgC ha⁻¹ per year, on average among the sites.





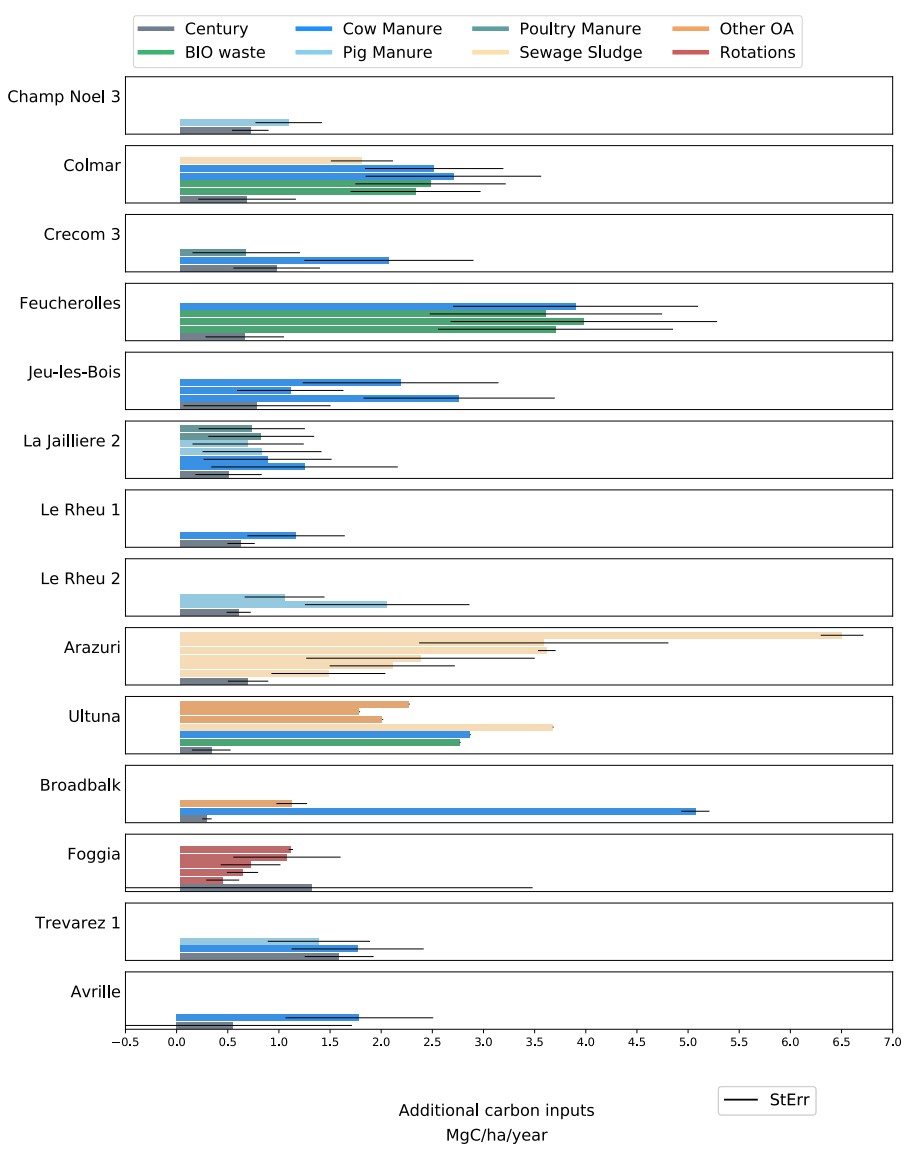

440

**Figure 8: Additional modelled carbon inputs (MgC ha$^{-1}$ year$^{-1}$) to reach the 4p1000 (grey bars) compared to additional carbon input treatments (colored bars) on each experimental site. Additional carbon inputs for field trials are calculated as the sum of organic fertilizers and the delta carbon inputs from crop yields (compared to the control plot). Additional carbon treatments are separated into different categories: BIO waste = biowaste compost, green manure, green manure + sewage sludge and household waste, Cow Manure = cow manure and farmyard manure (in *Broadbalk* and *Ultuna*), Pig Manure, Poultry Manure, Sewage Sludge, Rotations = different crop rotations, Other organic amendments (OA) = straw, sawdust and peat (in *Ultuna*) and Castor Meal (in *Broadbalk*). The error bars shown are the standard errors computed with the Monte Carlo method.**



### 3.3. Carbon inputs change in future scenarios of temperature increase

The temperature sensitivity analysis of the Century model for the 4p1000 target framework is plotted in Fig. 9. The required amount of C inputs to reach the 4p1000 target is likely to increase with increasing temperature scenarios. In particular, carbon inputs will have to increase on average by 54% in the AS1 scenario of +1°C and 120% in the AS5 scenario of +5°C temperature change. This represents an additional C inputs increase of 11% and 77% respectively, compared to the business as usual situation with current temperature setup. What can be clearly seen in the graph is the increased amount of C inputs required in *Trévarez*, where C inputs should more than quadruplicate to reach the 4p1000 objective.

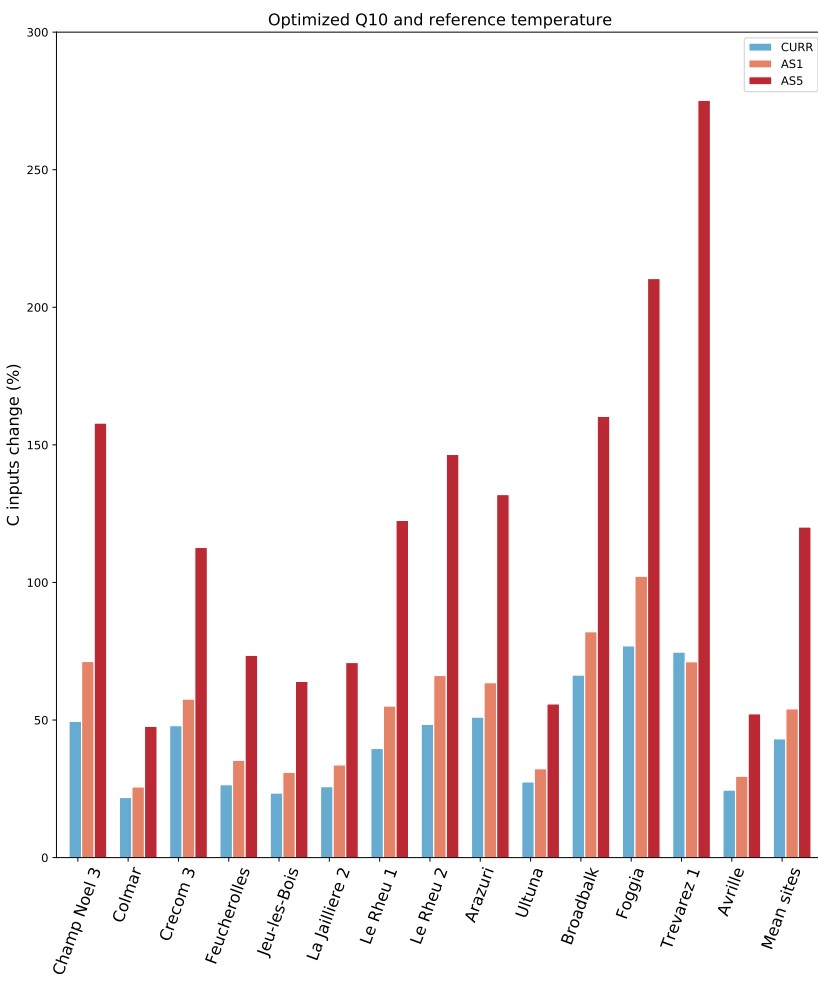

**Figure 9: Temperature sensitivity analysis of carbon inputs change (%) to reach the 4p1000 objective. CURR=business as usual simulation, AS1=RCP2.6 scenario of +1°C temperature increase, AS5=RCP8.5 scenario of +5°C temperature change.**



## 4 Discussion

### 4.1. Reliability of the Century model

The Century model has been widely used to simulate SOC stocks dynamics in arable cropping systems (Bortolon et al., 2011; Cong et al., 2014; Kelly et al., 1997; Xu et al., 2011). Optimizing the metabolic:structural ratio in the reference plots allowed us to initialize the carbon inputs compartments, since no measurement of the lignin:nitrogen ratio was available. This allowed us: 1) taking into account the average carbon quality of the litter pools in the different crops rotations and 2) correctly estimating the initial values of SOC stocks on the majority of the sites. On the other side, this could have influenced the predicted redistribution of C in the additional C inputs required to reach the 4p1000 (Fig. 5). We suggest that taking into account the historical site-specific land use could help initializing SOC stocks without requiring any assumption on the metabolic:structural ratio (e.g. with historically based equilibrium scenarios as in Lugato et al. (2014)). To further improve SOC stock simulations, we decided to optimize the $Q_{10}$ and reference temperature parameters on the reference plots, to account for the different pedo-climatic conditions of the experimental sites and enhance model predictions of SOC stocks dynamics (Craine et al., 2010; Lefèvre et al., 2014; Meyer et al., 2018; Wang et al., 2010). Although the dispersion of SOC stocks over time is not perfectly captured in the majority of the control plots (see the high LC component of the MSD in Fig. 4), the simulations of SOC dynamics were improved by the optimization of temperature related parameters and the NRMSD was found to be lower than 15% on all sites. However, the capability of Century to simulate SOC stocks variation on the virtual simulations of additional C treatments might be a major shortcoming of modeling results. In fact, although SOC stocks were found to be increasing on average in the additional C treatments (0.25% per year with 1.52 MgC ha$^{-1}$ yearly additional carbon inputs), this increase rate is lower than the 0.4% increase of SOC stocks predicted by Century with lower amounts of virtual C inputs (0.66 MgC ha$^{-1}$ per year).

### 4.2. Increasing annual SOC stocks by 4p1000

#### 4.2.1. Modelled carbon inputs to reach the 4p1000

Century simulations estimated that annual carbon inputs should increase by 43±5% (SE) on average to reach the 4p1000 target on the selected experimental sites, under the condition that the additional carbon inputs are equally distributed among the surface and belowground, in order to maintain the same aboveground:belowground ratio as at the beginning of the experiment. This is higher than the values found by Chenu et al. (2019) using default RothC 26.3 parameters, who estimated a relative increase of C inputs in temperate sandy soils by 24% and in temperate clayey soils by 29%. However, not only the quantity of carbon but also the quality will need to change according to Century predictions. In fact, the predicted aboveground structural litter change was threefold higher than all other pools on average, representing an additional 0.18 MgC ha$^{-1}$ each year. A way for the farmer to increase the structural fraction of the carbon inputs is to compost the organic amendments that will be spread on soil surface. Increasing EOM in large quantities may not be





possible everywhere. First of all, the amount of organic fertilizers is limited at site scale and farmers may
have difficulties in producing or buying high quantities of EOMs (Poulton et al., 2018). Secondly, farmers
may be prevented from applying high amounts of EOM because of the risk of nitrate and phosphate pollution
(Li et al., 2017; Piovesan et al., 2009).

#### 4.2.2. Stability of the additional carbon stored

Another important aspect to take into consideration is the stability of the additional carbon. In fact, the
duration and persistence of carbon in the soil might be very different whether or not the proportion of stable
carbon is important. In the Century model, this translates into questioning whether the fractions of the long
turnover rate pools (the slow and passive SOC pools) have increased. In our simulations, a general pattern
can be detected (Fig. 6) where both passive and slow pools increased, but at very different rates (0.1‰ and
6.1‰ per year respectively). The active pool increased by 5.8‰ annually, with benefits for soil fertility and
hence food security. The additional carbon is essentially slow (2.7 MgC ha$^{-1}$ in 30 years of simulations),
meaning that it will be stored in the soil for around 20 to 30 years. The increase in carbon inputs must be
sustained to increase SOC stocks at the desired rate, until a new equilibrium will be reached. To further
increase SOC stocks after the new equilibrium, we might consider implementing new strategies of additional
carbon later on. For instance, this could be achieved through the implementation of complementary
management options to those considered in the long-term experiments described here, such as residues
management, cover crops, conservation agriculture and agroforestry systems (Chenu et al., 2019; Lal, 1997;
Smith et al., 1997).

#### 4.2.3. Simulated carbon inputs and experimental carbon addition treatments

Different types of organic carbon treatments were considered in this study and compared to Century
simulations of carbon inputs required to reach the 4p1000. In all experimental sites with additional EOM
inputs, at least one treatment employed higher amounts of C inputs compared to the simulated C inputs
required for a 4‰ annual target. In *Foggia*, carbon inputs from different crop rotations were studied, but
none employed sufficient amounts of additional carbon to reach the 4p1000, as predicted by Century. Model
results in *Foggia* had a high standard error, mainly due to the fact that the variability of crop yields for this
site was not available. Thus, for this site, we calculated model uncertainty using the average relative
variability across the whole dataset, which could have increased the uncertainty of model outputs.

It is important to note that the amount of carbon inputs simulated by Century was constrained to have the
same aboveground:belowground ratio as at the beginning of the experiment. This means that the additional
carbon inputs should be distributed equally on soil surface and belowground, not to change the initial
allocation of carbon in the litter pools. Since all field treatments were performed under conventional tillage,
the comparison between modelled and observed additional carbon inputs under this constraint holds well.

The annual SOC stocks variation (0.25%) estimated in the experimental carbon treatments across the 14 sites,
indicates that Century might be overestimating the effect of additional carbon inputs on SOC stocks. In


particular, only 19 out of 46 field treatments (with average additional C inputs of 1.93 MgC ha$^{-1}$ per year)
were found to be actually increasing SOC stocks at a higher rate than 4‰ per year, relatively to their initial
SOC stocks. This is similar to the values found by Poulton et al. (2018), who estimated that adding similar
high amounts of C inputs increased SOC stocks at an annual rate higher than 4‰ in 16 long-term agricultural
experiments. The overestimation of the Century model might be due to several factors. First of all, the C
inputs prescribed to model simulations were constant through time, while C inputs from plant material
actually vary annually and over the years because of agronomical and climatic factors. Historical land use
and management practices such as tillage were not taken into account, although they affect SOC stocks
(Pellerin et al. 2017). Another factor that the model is not taking into account is nitrogen and other nutrients
availability, which might affect the SOC stocks dynamics. This is especially true in treatments with different
frequencies of application (e.g. *Arazuri*), where nutrients depletion is likely to be more evident when the
application is sparser. The calculation method of C inputs also influences the simulation of SOC stocks
(Clivot et al., 2019). However, estimating the increase of carbon inputs relatively to their initial value has
likely cancelled out uncertainties related to the C inputs estimation method in our analysis.
**4.2.4. Organic carbon inputs use in Europe**
Zhang et al. (2017) estimated that the proportion of nitrogen inputs from livestock manure applied to
European croplands was 3.9 Tg N in 2014, for a cropland area of 127 M ha in 2015 (Goldewijk et al. 2017).
Cattle manure, which represents the highest proportion of manure produced and applied to croplands, has
average C:N ratio ranging between 10 and 30 (multiple sources from Fuchs et al. (2014) and Pellerin et al.
(2017)). With these data, we can roughly estimate the application of C manure from livestock in European
agricultural soils as ranging between 0.30 and 0.92 MgC ha$^{-1}$ each year. Most of the experiments used in this
study used higher amounts of C inputs (1.52 MgC ha$^{-1}$ per year on average). However, the C inputs need
predicted by Century, which ranged between $0.24\pm0.02$ and $1.20\pm1.00$ MgC ha$^{-1}$ per year, plus one site with
$1.45\pm0.16$ MgC ha$^{-1}$ per year, is in line with the average use of livestock manure in Europe. In terms of C
sequestration, organic fertilizers coming from animal manure are usually being applied to the soil at some
location, hence they cannot account for additional climate mitigation potential (Poulton et al., 2018).
However, according to Zhang et al. (2017) estimation, there is room for improvement since the fraction of
livestock manure applied to cropland in the 2010s was approximately 26% of total livestock production in
Europe. The estimates from Zhang et al. (2017) refer to livestock manure only. In our study, we also
considered treatments with other types of EOM addition, such as sewage sludge and household waste. These
should be accounted for as they represent additional C inputs to agricultural soils. Moreover, in many
countries a significant proportion of food and urban waste is currently left on disposal areas, where carbon is
lost to the atmosphere as $CO_2$ or $CH_4$ emissions (Bijaya et al. 2006). Total sewage sludge used in Europe
(EU26) for agriculture can be calculated from Eurostat (2014b) as $4558 \cdot 103$ MgDM per year (in 2010).
Using the Van Bemmelen factor (1.724) to convert OM to OC (McBratney and Minasny, 2010; Rovira et al.,
2015), we can estimate the sewage sludge used in European croplands as being around 0.021 MgC ha$^{-1}$ per



year. Moreover, Pellegrini et al. (2016) found that sewage sludge reuse in agriculture is increasing in Europe.
In 2018, household waste composted in Europe (EU27) was 37M MgDM (Eurostat, 2020). Considering a
carbon content in household waste of 71% (Larsen et al., 2013) and assuming that all and only composted
household waste is used in agriculture, we can approximate household waste use in Europe as being 0.2 Mg
C ha$^{-1}$ per year. A contribution to the sequestration of C from the atmosphere could also come from changing
the treatment methods which affect the quality of C in crop residues and manure, so that their turnover time
increases, e.g. through fermentation or biochar. In general, improving the use efficiency of EOM to the soil
by managing it differently could contribute to some extent to climate change mitigation, increase soil quality,
and reduce mineral fertilizers use (Chadwick et al. 2015).
**4.2.5. Reaching a 4p1000 target: only a matter of initial SOC stocks?**
As we could expect, the estimated amount of carbon inputs to reach the 4p1000 target was linearly correlated
to the initial observed level of SOC stocks (Fig. 7). This is primarily due to the linear structure of the Century
model. In fact, if we consider the stationary solution for which Eq. (2) is equal to 0, SOC depends linearly
on the carbon inputs. Therefore, the opposite is also true (i.e. carbon inputs are linearly dependent to the
initial amount of SOC stocks). Moreover, the 4p1000 target itself is defined as the increase of SOC by 0.4%
per year, relatively to its initial value (Minasny et al., 2017). Hence, it implies a proportional contribution
that depends on the initial SOC stocks. Wiesmeier et al. (2016) also observed a linear relationship between
SOC increase and C inputs. This linear relationship means that soils with high SOC stocks will have to
increase their carbon stocks more in absolute terms to meet this quantitative target. On the other side, smaller
amounts of C will have to be employed in sites with low levels of SOC stocks, to reach a 4p1000 target.
However, increasing C inputs where SOC stocks are low might require substantial changes in the agricultural
systems and such quantity of additional OM might not be available at a large scale. A counterpoint is also
that the 4p1000 initiative needs all the soils to increase their SOC stocks by 4‰ per year, even those with
medium or high SOC stocks (i.e. higher than 50 MgC ha$^{-1}$, such as grasslands and forests), where the required
additional C increase will be higher according to Century. This result depends on the quality of the simulated
carbon inputs (i.e. the predicted metabolic:structural ratio) and does not take into account any notion of soil
saturation.  Before applying this trend to calculate the required C inputs from current SOC stocks, we should
extend the database to cover different pedo-climatic regions of the word and use a multi-model analysis to
cut out individual model uncertainty.
**4.3.  Sensitivity analysis**
The predicted need of additional C inputs to reach the 4p1000 target is likely to be higher with future global
warming, as a consequence of modified SOC decomposition rates. Considering the crucial role of soil as a
land-use based option for mitigating climate change, recent studies have shown a growing interest in
temperature sensitivity of SOC stocks decomposition (Dash et al., 2019; Koven et al., 2011; Parihar et al.,
2019; Wiesmeier et al., 2016). We know that a significant fraction of SOM is subject to increasing



decomposition due to temperature sensitivity. However, the magnitude of expected feedbacks from SOC
stocks is still surrounded of controversy. In particular, this is mainly due to the diversity of organic
compounds in the soil that are known to have inherent sensitivities to temperature (Davidson and Janssens,
2006). In this context, the study of the Century model response to predicted scenarios of temperature increase
is of primary importance. We mimicked the most optimistic (+1˚C) and pessimistic (+5˚C) RCPs scenarios
of the 5th IPCC assessment report. What is striking from our results is that with increasing temperatures all
sites will have to provide considerably higher amounts of C inputs to reach the 4p1000 target (Fig. 9). In
particular, the C inputs change needs to more than double in all sites, according to the worst-case scenario of
+5˚C. It is important to point out that the optimization of the $Q_{10}$ and reference temperature parameters are
likely to influence the outcomes of the simulated SOC stocks and therefore the C inputs need. Nevertheless,
comparing the carbon input change simulated with the optimized version of Century (Fig. 9) to that simulated
with the default parameters setting (Fig. C1), shows that the predicted inputs change follows the same pattern,
even though the intensity of the increase is considerably higher in the optimized version. These results can
be understood in two ways. Either the optimized version of Century is overestimating the effect of
temperature on SOC stocks decomposition, or SOC stocks decomposition patterns are likely to increase even
more intensively when considering the entire range of possible $Q_{10}$ values. In either case, further research is
needed to reduce the uncertainty around the impact of climate change on SOC decomposition. Studies should
also examine moisture change, which we did not take into account here. This is likely to be impacted as a
consequence of modified precipitations and temperature (IPCC, 2015). Additionally, increased temperature
and $CO_2$ concentration in the atmosphere, as well as changes in precipitations are likely to influence net
primary production and therefore C inputs to the soil. All these feedbacks are important and must be taken
into account for a comprehensive evaluation of carbon cycle effects on climate change.
**5  Conclusion**
The Century model predicted an average increase of annual carbon inputs by 43±5% to reach a 4p1000 target
over a range of 14 agricultural sites across Europe, with diverse soil types, climates, crop rotations and
practices. The required simulated amount of additional C inputs was found to be systematically lower or
similar to the 46 treatments of carbon inputs carried out in these sites. However, Century might be
overestimating the predicted effect of additional C inputs on the SOC stocks variation rate, as the only field
treatments that were found increasing SOC stocks by at least 4‰ annually were those using very high
amounts of C inputs (~1.93 MgC ha$^{-1}$ per year). The predicted amount of additional carbon inputs depended
linearly on the initial amount of observed SOC stocks in the control experiments, indicating that lower
amounts of carbon inputs might be sufficient to reach the 4p1000 target where SOC stocks are low. However,
increasing C inputs might require substantial changes in the agricultural systems and high quantities of
additional organic matter might not be available at a large scale. The required amount of additional C inputs
was found to substantially increase with future scenarios of changes in temperature, rising concern on the
feasibility of a 4p1000 target under climate change and beyond that, the feasibility of SOC stocks



preservation. Promoting and applying soil carbon conservation strategies, namely redistributing crop residues
and organic amendments to the soil, implementing cover crops and conservation agriculture, developing
agroforestry and diversifying crop rotations, improves soil fertility and food production. The magnitude of
SOC storage potential in agricultural soils largely depends on site-specific conditions, such as climate, soil
type and land use. In this study, we only considered temperate, sub-humid and Mediterranean climates. A
broader evaluation of the required carbon inputs and associated agricultural practices to increase SOC stocks
is worthwhile to be carried out at larger scales. We also suggest that future research focuses on multi-
modeling analysis, to allow for a correct estimation of the uncertainties related to model-specific
assumptions.
**Authors contribution**
YH provided the initial model code. EB edited and developed the model code, performed the simulations
and prepared the manuscript with contributions from all co-authors. HC, IV, RF, TK and MM provided the
data.
**Competing interests**
The authors declare that they have no conflict of interest.
**Acknowledgements**
This work benefited from the French state aid managed by the ANR under the "Investissements d'avenir"
programme with the reference ANR-16-CONV-0003 (CLAND project). We acknowledge Mancomunidad
de la Comarca de Pamplona for maintenance and access to Arazuri site data. Research grant RTA2017-
00088-C03-01 form the Instituto Nacional de Investigación Agraria y Alimentaria, INIA (Spanish Agency).
We acknowledge Margaret Glendining, curator of the electronic Rothamsted Archive (e-RA) for providing
the Broadbalk data.
**Appendix A – Century model description and environmental functions used**
The temporal evolution of soil organic carbon is described in the Century model as a first order differential
matrix equation:
$$\frac{\mathrm{d}\boldsymbol{SOC}(\mathrm{t})}{\mathrm{dt}} = \boldsymbol{I} + \mathbf{A} \cdot \boldsymbol{\xi}_{\mathbf{TWLCI}}(\mathrm{t}) \cdot \mathbf{K} \cdot \boldsymbol{SOC}(\mathrm{t}),$$ (2)
where $\boldsymbol{SOC}(t)$ is the vector describing the SOC state variables. The first term on the right side of the equation
represents carbon inputs to the soil coming from plant residues and organic material. Carbon inputs are
allocated into four different litter pools. Hence, $\boldsymbol{I}$ is a 1x7 matrix with four nonzero elements. The second
term of the equation represents carbon outputs from the soil, following a first order decay kinetics. $\mathbf{A}$ is a 7x7



carbon transfer matrix that quantifies the transfers of carbon among the different pools. The diagonal entries
of **A** are equal to -1, denoting the entire decomposition flux that leaves each carbon pool. The non-diagonal
elements represent the fraction of carbon that is transferred from one pool to another. **K** is a 7x7 diagonal
matrix with the diagonal elements representing the potential decomposition rate of each carbon pool.
$\boldsymbol{\xi_{TWLCI}}(t)$ is the environmental scalar matrix, a 7x7 diagonal matrix with each diagonal element denoting
temperature $(f_T(t))$, water $(f_W(t))$ lignin $(f_{L\,i})$ and clay $(f_{Clay\,i})$ scalars, which modify the potential
decomposition rate. Temperature response function $f_T(t)$ is described by Eq. (4), the others are expressed as
follows. The moisture function $f_W(t)$ is a polynomial function ranging from 0.25 and 1 and taking the form
of:
$f_W(t) = -1.1 \cdot w^2 + 2.4 \cdot w - 0.29,$       (A1)
where $w$ is the daily relative humidity $(m^3{}_{water}\, m^{-3}{}_{soil})$.
The decomposition rate of structural litter pools is affected by their lignin content:
$f_{L\,i} = e^{-lgc \cdot L},$       (A2)
where $lgc$ is the coefficient that regulates the lignin effect, while $L$ is the lignin structural fraction of the
aboveground and the belowground litter pools.
Finally, the fraction of clay in the soil $(g\ clay\ g^{-1}soil)$ influences the decomposition rate of the active pool:
$f_{Clay\,i} = 1 - 0.75 \cdot clay.$       (A3)
**Appendix B – Model evaluation**
Two residual-based metrics were used to evaluate the goodness-of-fit of modeled and observed SOC stocks
for each site: the Mean Squared Deviation (MSD) and the Normalized Root Mean Squared Deviation
(NRMSD). The MSD for each site is defined as:
$MSD = \frac{\sum_{i=1}^{n}(m_i - o_i)^2}{s},$       (B1)
where $i = 1,...,n$ is the year of the experiment, $m_i$ and $o_i$ are respectively modeled and observed values of
SOC stocks and s is the number of observations in the experiment. Following Gauch et al. (2003), the MSD
can be decomposed into three components: the Squared Bias (SB), the Non-Unity slope (NU) and the Lack
of Correlation (LC). SB is calculated as:
$SB = (\bar{m} - \bar{o})^2,$       (B2)
where $\bar{m}$ and $\bar{o}$ are the mean values of modeled and observed SOC stocks respectively.
Calling $\Delta M_i = (\bar{m} - m_i)$ and $\Delta O_i = (\bar{o} - o_i)$ we have:
$NU = \left(1 - \frac{\sum_{i=1}^{n} \Delta M_i \cdot \Delta O_i}{\sum_{i=1}^{n} \Delta M_i^2}\right)^2 \cdot \frac{\sum_{i=1}^{n} \Delta M_i^2}{s},$       (B3)
$LC = \left(1 - \frac{\sum_{i=1}^{n}(\Delta M_i \cdot \Delta O_i)^2}{\sum_{i=1}^{n} \Delta O_i^2 \cdot \sum_{i=1}^{n} \Delta M_i^2}\right) \cdot \frac{\sum_{i=1}^{n} \Delta O_i^2}{s}.$       (B4)
These three components add up to MSD and help locating the causes of error of model predictions,
determining areas in the model that require further improvement (Bellocchi et al., 2010). In particular, SB





provides information about the mean bias of the simulation from measurements, NU indicates the capacity
of the model to correctly reproduce the magnitude of the fluctuation among the measurements and LC is an
indication of the dispersion of the points over a scatterplot, i.e. the capacity of the model to reproduce the
shape of the data (Kobayashi and Salam, 2000).
The second statistical measure we used was computed as the squared root of the MSD, normalized by the
mean observed SOC stocks:
$NRMSD = \frac{\sqrt{MSD}}{\bar{o}} \cdot 100.$         (B5)
This indicator is expressed as a percentage and allows to evaluate the model performance independently to
the units of SOC stocks.
**Appendix C – Sensitivity analysis with default Century parameters**

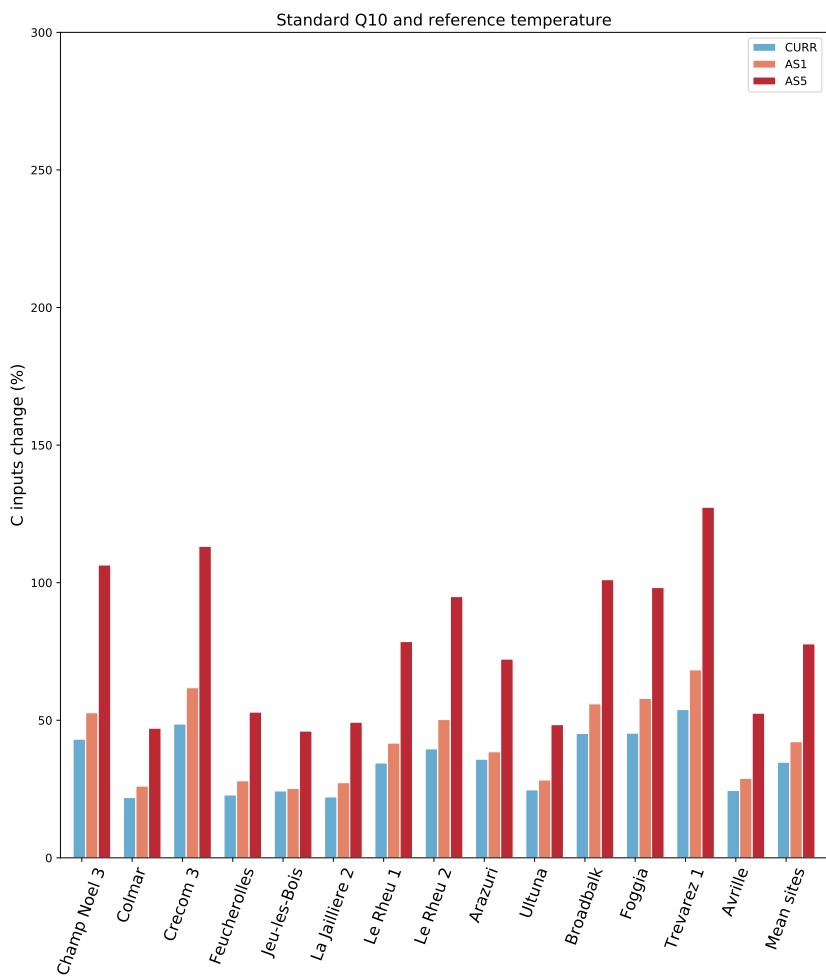






**Figure C1: Temperature sensitivity analysis of carbon inputs change (%) to reach the 4p1000 objective, using**
**Century default Q10 and reference temperature parameters. CURR=business as usual simulation, AS1=RCP2.6**
**scenario of +1˚C temperature increase, AS5=RCP8.5 scenario of +5˚C temperature change.**

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

Methods in Determining Soil Organic Carbon Sequestration Rates, Soil Science Society of America
Journal, 78(2), 348–360, doi:10.2136/sssaj2013.09.0412, 2014.
Pachauri, R. K., Mayer, L. and Intergovernmental Panel on Climate Change, Eds.: Climate change 2014:
synthesis report, Intergovernmental Panel on Climate Change, Geneva, Switzerland., 2015.
Parton, W. J., Stewart, J. W. B. and Cole, C. V.: Dynamics of C, N, P and S in grassland soils: a model,
Biogeochemistry, 5(1), 109–131, doi:10.1007/BF02180320, 1988.
Parton, W. J., Scurlock, J. M. O., Ojima, D. S., Gilmanov, T. G., Scholes, R. J., Schimel, D. S., Kirchner, T.,
Menaut, J.-C., Seastedt, T., Garcia Moya, E., Kamnalrut, A. and Kinyamario, J. I.: Observations and
modeling of biomass and soil organic matter dynamics for the grassland biome worldwide, Global
Biogeochem. Cycles, 7(4), 785–809, doi:10.1029/93GB02042, 1993.
Paustian, K., Lehmann, J., Ogle, S., Reay, D., Robertson, G. P. and Smith, P.: Climate-smart soils, Nature,
532(7597), 49–57, doi:10.1038/nature17174, 2016.

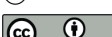



Pellegrini, M., Saccani, C., Bianchini, A. and Bonfiglioli, L.: Sewage sludge management in Europe: a
critical analysis of data quality, IJEWM, 18(3), 226, doi:10.1504/IJEWM.2016.10001645, 2016.
Pellerin, S., Bamière, L., Denis, A., Béline, F., Benoit, M., Butault, J.-P., et al.: Stocker du Carbone dans les
sols Français - Quel Potentiel au Regard de L'objectif 4 pour 1000 et à Quel Coût? Synthèse du
rapport d'étude. ADEME., Environ. Sci. Policy, 77, 130–139, doi:doi:
10.1016/j.envsci.2017.08.003, 2017.
Piovesan, R. P., Favaretto, N., Pauletti, V., Motta, A. C. V. and Reissmann, C. B.: Perdas de nutrientes via
subsuperfície em colunas de solo sob fertilização mineral e orgânica, Rev. Bras. Ciênc. Solo, 33(4),
757–766, doi:10.1590/S0100-06832009000400002, 2009.
Poulton, P., Johnston, J., Macdonald, A., White, R. and Powlson, D.: Major limitations to achieving "4 per
1000" increases in soil organic carbon stock in temperate regions: Evidence from long-term
experiments at Rothamsted Research, United Kingdom, Glob Change Biol, 24(6), 2563–2584,
doi:10.1111/gcb.14066, 2018.
Powlson, D. S., W., A. P.: The potential to increase soil carbon stocks through reduced tillage or organic
material additions in England and Wales: A case study., Agriculture, Ecosystems and Environment,
146, 23–33, doi:doi:10.1016/j.agee.2011.10.004, 2012.
Redin, M., Recous, S., Aita, C., Dietrich, G., Skolaude, A. C., Ludke, W. H., Schmatz, R. and Giacomini, S.
J.: How the chemical composition and heterogeneity of crop residue mixtures decomposing at the
soil surface affects C and N mineralization, Soil Biology and Biochemistry, 78, 65–75,
doi:10.1016/j.soilbio.2014.07.014, 2014.
Rovira, P., Sauras, T., Salgado, J. and Merino, A.: Towards sound comparisons of soil carbon stocks: A
proposal based on the cumulative coordinates approach, CATENA, 133, 420–431,
doi:10.1016/j.catena.2015.05.020, 2015.
Saffih-Hdadi, K. and Mary, B.: Modeling consequences of straw residues export on soil organic carbon, Soil
Biology and Biochemistry, 40(3), 594–607, doi:10.1016/j.soilbio.2007.08.022, 2008.
Sanderman, J., Hengl, T. and Fiske, G. J.: Soil carbon debt of 12,000 years of human land use, Proc Natl
Acad Sci USA, 114(36), 9575–9580, doi:10.1073/pnas.1706103114, 2017.
Smith, P., Powlson, D., Glendining, M. and Smith, J.: Potential for carbon sequestration in European soils:
preliminary estimates for five scenarios using results from long-term experiments, Global Change
Biology, 3(1), 67–79, doi:10.1046/j.1365-2486.1997.00055.x, 1997.
Soussana, J.-F.: Matching policy and science_ Rationale for the '4 per 1000 - soils for food security and
climate' initiative, 14, 2017.
VandenBygaart, A. J.: Comments on soil carbon 4 per mille by Minasny et al. 2017, Geoderma, 309, 113–
114, doi:10.1016/j.geoderma.2017.05.024, 2018.
Wang, X., Piao, S., Ciais, P., Janssens, I. A., Reichstein, M., Peng, S. and Wang, T.: Are ecological gradients
in seasonal Q10 of soil respiration explained by climate or by vegetation seasonality?, Soil Biology
and Biochemistry, 42(10), 1728–1734, doi:10.1016/j.soilbio.2010.06.008, 2010.
Wiesmeier, M., Poeplau, C., Sierra, C. A., Maier, H., Frühauf, C., Hübner, R., Kühnel, A., Spörlein, P., Geuß,
U., Hangen, E., Schilling, B., von Lützow, M. and Kögel-Knabner, I.: Projected loss of soil organic
carbon in temperate agricultural soils in the 21st century: effects of climate change and carbon input
trends, Sci Rep, 6(1), 32525, doi:10.1038/srep32525, 2016.
Wollenberg, E., Richards, M., Smith, P., Havlík, P., Obersteiner, M., Tubiello, F. N., Herold, M., Gerber, P.,
Carter, S., Reisinger, A., van Vuuren, D. P., Dickie, A., Neufeldt, H., Sander, B. O., Wassmann, R.,
Sommer, R., Amonette, J. E., Falcucci, A., Herrero, M., Opio, C., Roman-Cuesta, R. M., Stehfest,
E., Westhoek, H., Ortiz-Monasterio, I., Sapkota, T., Rufino, M. C., Thornton, P. K., Verchot, L.,
West, P. C., Soussana, J.-F., Baedeker, T., Sadler, M., Vermeulen, S. and Campbell, B. M.:
Reducing emissions from agriculture to meet the 2 °C target, Glob Change Biol, 22(12), 3859–3864,
doi:10.1111/gcb.13340, 2016.
Xia, J. Y., Luo, Y. Q., Wang, Y.-P., Weng, E. S. and Hararuk, O.: A semi-analytical solution to accelerate
spin-up of a coupled carbon and nitrogen land model to steady state, Geosci. Model Dev., 5(5),
1259–1271, doi:10.5194/gmd-5-1259-2012, 2012.
Xu, W., Chen, X., Luo, G. and Lin, Q.: Using the CENTURY model to assess the impact of land reclamation
and management practices in oasis agriculture on the dynamics of soil organic carbon in the arid
region of North-western China, Ecological Complexity, 8(1), 30–37,
doi:10.1016/j.ecocom.2010.11.003, 2011.



Zhang, B., Tian, H., Lu, C., Dangal, S. R. S., Yang, J. and Pan, S.: Global manure nitrogen production and
928          application in cropland during 1860–2014: a 5 arcmin gridded global dataset for Earth system
929          modeling, Earth Syst. Sci. Data, 9(2), 667–678, doi:10.5194/essd-9-667-2017, 2017.
Zinn, Y. L., Lal, R. and Resck, D. V. S.: Changes in soil organic carbon stocks under agriculture in Brazil,
931          Soil and Tillage Research, 84(1), 28–40, doi:10.1016/j.still.2004.08.007, 2005.