# Peer review of "Table S1: Default parameters of the Century model affecting litter and SOC dynamics (Parton et al., 1988)."

_Biogeosciences, 2020_

## Author Response (AR2)

**Referee 1**

General comments

Overall, the manuscript entitled 'Additional carbon inputs to reach a 4 per 1000 objective in Europe: feasibility and projected impacts of climate change based on Century simulations of long-term arable experiments' will be probably of great interest for the readers of Biogeosciences Journal. Indeed, it provides interesting results on additional carbon inputs that would be necessary to reach the 4 per 1000 objective in long-term arable experiments from Europe. Also, it also highlights the impact of temperature increase on these additional carbon inputs. Overall the manuscript is well written, the paragraphs are well organized and the ideas are well supported by relevant references. It was also appreciated that most of the limits of the approach have been underlined. That's why I think that this manuscript should be considered for publication but I have noticed some points that need to be addressed before publication.

**We thank the reviewer for the positive comments. We addressed the issues raised by modifying some parts of the manuscript. Please find the details below.**

Detailed comments

In the next paragraph, I developed these points and made some comments that will help the authors to improve the manuscript.

L.33

Overall, I think that the abstract should be improved. It seems to me that it did not reflect the good quality of the manuscript. The authors should clearly highlight the objectives of the study.

**We rephrased some sentences in the abstract, to better clarify the objectives of the study.**

- **L.33-36 were rephrased as follows:**
**"In this study, we assessed the amount of organic C inputs that are necessary to reach a target of SOC stocks increase by 4‰ per year on average, for 30 years, in 14 long-term agricultural sites in Europe. We used the Century model to simulate SOC stocks and assessed the required level of additional C inputs to reach the 4 per 1000 target in these sites. Then, we analyzed how this would change under future scenarios of temperature increase."**

- **L.46-51 were rephrased as follows:**
**"In the experimental sites, we found that SOC stocks in treatments with additional C inputs were increasing by 0.25% on average. This means that the C inputs required to reach the 4 per 1000 target might actually be much higher. Furthermore, we estimated that annual C inputs will have to increase even more due to climate warming, that is 54% more and 120% more, for a 1°C and 5°C warming, respectively. We showed that modeled C inputs required to reach the target depended linearly on the initial SOC stocks, raising concern on the feasibility of the 4 per 1000 target in soils with a higher potential contribution on C**

**sequestration, that is soils with high SOC stocks. Our work highlights the challenge of increasing SOC stocks at large scale and in a future with warmer climate."**

L.35-36

After reading the title and the first sentences of the abstract, it is not clear if the required level of carbon inputs is assessed only for the long-term agricultural experiments or for other soils? After reading the entire manuscript, we understand that the model simulation concerns only the long-term experiments. But it should be better underlined in the abstract.

**We rephrased two sentences in the abstract, to better specify that the objective of our modeling exercise is to estimate the carbon inputs in the 14 long-term experiments, as follows:**
**(L.33-36)**
**"In this study, we assessed the amount of organic C inputs that are necessary to reach a target of SOC stocks increase by 4‰ per year on average, for 30 years, in 14 long-term agricultural sites in Europe. We used the Century model to simulate SOC stocks and assessed the required level of C inputs increase to reach the 4 per 1000 target in these sites."**

**In the introduction, we also rephrased the part stating the objectives of our work, to take into account this comment:**
**"Our work was set up in this context with the objectives to: 1) estimate the amount of C inputs needed to increase SOC stocks by 4‰ per year; 2) investigate if this amount is attainable with currently implemented soil practices (i.e. organic amendments and different crop rotations) and 3) study how the required C inputs are going to evolve in a future driven by climate change. We used the biogeochemistry SOC model Century, which is one of the most widely used and validated models (Smith et al., 1997) to simulate SOC stocks in 14 different agricultural LTEs around Europe. We set the target of SOC stocks increase to 4‰ per year for 30 years, relative to the initial stocks in the reference treatments. With an inverse modeling approach, we estimated the amount of additional C inputs required to reach a 4p1000 target at these sites. Finally, we evaluated the dependency of the required additional C inputs to different scenarios of increased temperature."**

L.39

The objective of determining the impact of temperature on the simulated additional carbon inputs to reach the 4 per 1000 objective should be highlighted in the objectives.

**We moved L.39 ("Then, we analyzed how this would change under future scenarios of temperature increase.") to L.37, when stating the objectives of the model.**

L.42-43

In this sentence, I think that it is not clear to understand if the additional C inputs are simulated from the current soil C stocks (which implies that there were C inputs in the previous years) or from the initial soil C stocks at the beginning of long-term experiments.

**We added to the sentence that the additional carbon inputs increase is calculated with respect to the initial carbon inputs in the control treatment:**

**"We found that, on average among the selected experimental sites, annual C inputs will have to increase by 43.15 $\pm$ 5.05 %, which is 0.66 $\pm$ 0.23 MgC ha$^{-1}$ per year (mean $\pm$ standard error), with respect to the initial C inputs in the control treatment."**

L.51

I would suggest to add a few sentences to conclude on the feasibility of reaching the 4 per 1000 objective. At the end of this abstract, the reader can wonder what are the conclusions and perspectives of this study.

**To express our concern on the feasibility of the 4 per 1000 highlighted by our work, we added the following sentence to L.50: "raising concern on the feasibility of the 4 per 1000 target in soils with a higher potential contribution on C sequestration, that is soils with high SOC stocks."**

**We also added the following sentence to the end of the abstract: "Our work highlights the challenge of increasing SOC stocks at large scale and in a future with warmer climate."**

L.87

See also Powlson et al. (2011) for the definition of soil C sequestration which implies a net removal of CO2 from the atmosphere.

Powlson, D.S., Whitmore, A.P., and Goulding, K.W.T. 2011. Soil carbon sequestration to mitigate climate change: a critical re-examination to identify the true and the false. European Journal of Soil Science 62:42-55.

**We added the cited reference.**

L.106

The authors should precise the reason of the initialization of the models. Is it because the initial SOC stocks are not available?

**We rephrased L.106-107 to explain that model initialization is required either for lack of data or to allocate carbon in the different model compartments, which cannot be measured, as follows:**

**"This means that the initial status of SOC has to be set, either for lack of data on total initial stocks, or to determine the allocation of C among model's compartments that cannot be measured. This is commonly accomplished by assuming that SOC is at equilibrium at the beginning of the experiment (Luo et al., 2017; Xia et al., 2012)."**

L.119

One sentence to explain the choice of this model?

**We added a sentence saying that Century is one of the most widely used and validated models and cited the following paper to support our statement:**

Smith, P., J.U. Smith, D.S. Powlson, W.B. McGill, J.R.M. Arah, O.G. Chertov, K. Coleman, et al. "A Comparison of the Performance of Nine Soil Organic Matter Models Using Datasets from Seven Long-Term Experiments." Geoderma 81, no. 1–2 (December 1997): 153–225. https://doi.org/10.1016/S0016-7061(97)00087-6.

**Also, we added the following sentence to the beginning of subsection 2.2.1:**

**"For this study, we selected the Century model, which has proved to be well suited to simulate accurately the soil dynamics in a range of pedoclimatic areas and cropping systems (Bortolon et al., 2011; Cong et al., 2014; Parton et al., 1993), and because we had the full command of the model for fine tuning of parameters."**

L.122

I think that it is important to remind that the simulations concern the long-term experiments and not other soils.

**We added: "at these sites" and rephrased two sentences to take into account other comments:**

**"We set the target of SOC stocks increase to 4‰ per year for 30 years, relative to the initial stocks in the reference treatments. With an inverse modeling approach, we estimated the amount of additional C inputs required to reach a 4p1000 target in these sites."**

L.127

There is one control plot for each long-term experiment but there are 14 control plots in total. So the sentence should be rephrased.

**We rephrased the sentence as follows:**

**"We compiled data from 14 LTEs in arable cropping systems across Europe (Fig. 1), where a total of 46 treatments with increased C inputs to the soil were performed and one control plot in each experiment was implemented."**

L.153-157
We need to have more details:

1/In Table 1, we have the carbon inputs for the crop rotations (so we can assess additional plant carbon inputs in comparison to the reference ). Are these additional plant carbon inputs included in the column of additional carbon inputs? If yes, that means that additional carbon inputs include both plant and manure inputs, right?

**Additional carbon inputs in the 6t$^h$ column of the table represent carbon inputs from the organic treatments only. Additional carbon inputs from different yields compared to the reference are included in column 4 (Carbon inputs from crop rotations). We reformulated the 6$^{th}$ column title and the table description to make it clearer, as follows: "Additional carbon inputs from organic treatment"**

**The only site where additional C inputs from different yields is equal to the additional organic treatment is Foggia. We clarified this by adding the following note: "*** In Foggia, additional carbon inputs from organic treatments were calculated for each rotation as the difference between C inputs in the rotation and the reference wheat-only rotation."**

2/ In the text, how do the authors appreciate if the 4 per 1000 objective is attainable? Do they assess the difference of SOC annual variation between treatments and reference?

**We consider that the 4 per 1000 objective is attained when the treatment reaches a 4‰ increase of SOC stocks, relative to the initial SOC stocks in the reference treatment. This is specified in L.327-331. However, we rephrase the following sentence in L.130-131 to clear it out earlier in the text:**

**"We set the target of SOC stocks increase to 4‰ per year for 30 years, relative to the initial stocks in the reference treatments."**

3/ Finally, I wonder if this section should not be part of the Results section? L.173-176

In Table 2, the presented initial SOC stocks seem to be measured (from the title). So I wonder why the initialization of the model was done by simulating initial SOC stocks. Why not using measured initial SOC stocks?

**Initial SOC stocks were measured. However, we needed to initialize the Century model to allocate SOC stocks at time 0 in the different model compartments. A relaxation approach to initialize the model such as in Dimassi et al. (2017) could not be used, due to the requirement of an equilibrium state to perform the optimization algorithm to calculate C inputs to reach a 4 per 1000 target.**

**For this reason, we consider that Table 2 is part of the material used, since it is only a description of the sites.**

L.79-181

I just wonder why details of sampling are described for this site but not for the other sites...

**We described Broadbalk separately because it did not have any replicates, like Foggia and Champ Noel. But unlike these sites, Broadbalk soil was sampled with a protocol allowing a better estimation of SOC concentration. We rephrase this sentence to make it clearer:**

**"SOC stocks were measured in 3 – 4 replicates, apart from Foggia and Champ Noël 3 experiments, where no replicates were available, and Broadbalk. In this experiment, SOC was measured in each plot using a semi-cylindrical auger where 10-20 cores were taken from across the plot and bulked together (more details can be found on the e-RA website[1]).**

L.194

As I said earlier, I think that the authors should add a few sentences somewhere to explain the choice of this model. Why this model instead of another one (ROTH C, DNDC...)?

**This was added in the introduction ("We used the biogeochemistry SOC model Century, which is one of the most widely used and validated models (Smith et al., 1997)").**
**Also, at the beginning of subsection 2.2.1, we added:**
**"For this study, we selected the Century model, which has proved to be well suited to simulate accurately the soil C dynamics in a range of pedoclimatic areas and cropping systems (Bortolon et al., 2011; Cong et al., 2014; Parton et al., 1993), and because we had the full command of the model for fine tuning of parameters. "**

L.235
The following points need to be clarified:

1/Do initial sizes correspond to initial SOC stocks at the beginning of the experiment or when SOC were measured after a certain number of years where we supposed an equilibrium?

**Initial sizes correspond to initial SOC stocks at the beginning of the experiment. We reformulate L.235 as follows to make it clear: "The initialization of the model consists in specifying the sizes of the SOC pools at the beginning of the experiment."**

2/If initial sizes of SOC refer to initial SOC stocks at the beginning of the long-term experiments, why not using measured initial SOC stocks?

**As explained earlier, although total SOC stocks are known, we needed to initialize the Century model to allocate SOC stocks at time 0 in the different model compartments. Since the Century compartments are conceptual their fraction can't be derived from observations.**

L.252
* * *
[1] www.era.rothamsted.ac.uk

I am not sure that this paragraph is at the proper place. I explain myself: the authors introduced the Century model in the previous paragraph and they go on with the Century model calibration in the following paragraph. In this paragraph, if I understand well, the C inputs are estimated by using allocation coefficients for each of the treatments of the long- term experiments, there are no direct use of the Century model. That's why I suggest to move this section.

**We moved this subsection to subsection 2.1.3, before the description of the Century model.**

L.366

As the calibration was partly done by using data from control, is it not normal to expect a good fit of modelled values to control SOC values, no? Why not checking the fit of calibrated model to the SOC values of the other treatments of the long-term experiments?

**Before choosing Q10 and reference temperature as the best parameters' calibration option, we tested different other parameters (i.e. the five pools decomposition parameters, Q10 only, reference temperature only etc.). In these calibration tests, model fit to observed SOC was not improved. This is why we decided to show how the parameters' optimization worked in the control plots. However, since a validation of SOC increase in the rest of the treatments is missing, we add Fig. 5 to paragraph 3.1 (see Fig_5.pdf in supplement). This figure shows how the virtual C inputs simulated by Century to reach the 4‰ target reproduce the correlation between additional C inputs and SOC stocks increase in the C inputs treatments.**

**As shown in Fig. 5, although the correlation of C inputs with SOC stocks increase is not clear-cut ($R^2=0.23$), with a large variability of the effect of C inputs on SOC stocks increase, we state that Century is generally overestimating the effect of additional C inputs on SOC stocks increase. We discuss this result stating that the hypothesis of equilibrium at the beginning of the simulation might be one of the major sources of this error.**

[Figure]

**[Figure 5: Correlation between additional carbon inputs (MgC ha⁻¹ per year) and annual SOC stock increase (%) in the carbon inputs treatments and mean ± standard deviation of the additional carbon inputs to reach the 0.4% target in Century.]**

L.410-412

But did the authors test the correlation between the optimal input increase and Q10 or decomposition rates?

**We tested it, but there was no significant correlation between optimal input increase and Q10 (see Fig_RC1.pdf in supplement)**

[Figure]

**[Fig.RC1: Correlation between total C inputs to reach the 4p1000 target (MgC ha⁻¹ per year) and optimized Q₁₀ parameter values]**

L.480-481

Where is this result (0.25% increase per year) presented? Also, I suggest the authors better introduce the additional C treatments (which are actually the real treatments in the long-term experiments) and the virtual treatments.

**The result is presented at L.438, now slightly reformulated ("In the experimental treatments were applied 1.52 MgC ha⁻¹ per year on average and SOC stocks were found to be increasing by 0.25% per year relative to initial stocks.")**

L.494

Just one comment: by composting the organic amendments that will be spread on soil surface, there will be some C emissions during the composting process so it will be necessary to make a full assessment of C cycle with and without composting to be sure that the composted C input result in net C sequestration.

**In this study we focused only on the soil C inputs, but we agree that a full assessment of the C cycle would be needed in a more exhaustive analysis (cf. following comment for line 573).**

L.496

Another comment: in the case of animal manure, if farmers produce more manure, it implies more animals and larger C emissions through animals. Consequently, even if more manure is returning to soil, it will not result in net C sequestration.

**We add these two comments to the end of subsection 4.2.1, with the following sentences:**

**"Moreover, producing additional animal manure implies larger GHG emissions through animal digestion and manure decomposition. Consequently, even if more manure is returned to the soil, it will not necessarily result in climate change mitigation."**

L.573-575
Are possible emissions through the different managements taken account?

**No, they are not. So we add a sentence to L. 573 to make it clear:**

**"However, a full C cycle assessment should be considered to make sure that GHG emissions associated to such treatments do not exceed additional C storage (Guenet et al., 2020)."**

L.589-591

I am not sure to understand that point. Do we really need that all soils increase their SOC stocks by 4 per 1000? Some soils could be increased by more than 4 per 1000 and if this counterbalances for other soils which cannot be increased by 4 per 1000, overall the objective should be attainable, no?

**We rephrase these sentences to make our point clearer:**
**"A counterpoint is also that the largest contribution of C sequestration will come from soils with medium or high SOC stocks (i.e. higher than 50 MgC ha$^{-1}$, such as grasslands and forests). In these soils, the required additional C inputs will have to be higher according to Century, raising concern on a compensation of $CO_2$ emissions through improved SOC stocks at a global scale"**

**Referee 2**

This study used the CENTURY model to simulate the C inputs required for the 4p1000 target to be met, for 14 long-term experiments. Climate data from the past 30 years were used for the simulations. The main findings suggest that the 4p1000 target can be reached with modest increases of exogenous organic matter (EOM) inputs (40-50 %). For two reasons however, this is probably a large underestimate. Firstly, comparison of the SOC changes in the 14 long-term experiments suggest that much larger EOM additions are required to reach the 4p1000 target than suggested by the 'virtual' simulations. Secondly, simulations under warmer conditions show that higher EOM additions will be required in the future than is indicated in the main simulations, in order to compensate for temperature-induced decreases in C stocks.

**General comments:**

The study is very relevant to current-day topics of interest and it will therefore be of interest to readers of Biogeosciences. The manuscript is generally well written and seems to have been written very transparently (one minor exception, see point 2 below). The introduction is interesting and the breadth of relevant subjects covered in the introduction and discussion is appropriate. The simulations have been carried out thoroughly. My biggest issue with the study is that the results are biased because of the two reasons (given above). Although the authors are honest about the biases in the Results section, the meaning of these biases needs to be communicated more clearly. At the moment, they are communicated adequately in the Discussion but not the Abstract. My suggestion to remedy this is given in point 3 below.

**We thank the reviewer for the positive comment and we modified the abstract to better show the limits of the study.**
**The end of the abstract was reformulated as follows: "This means that the C inputs required to reach the 4 per 1000 target might actually be much higher. Furthermore, we estimated that annual C inputs will have to increase even more due to climate warming, that is 54% more and 120% more, for a 1°C and 5°C warming, respectively. We showed that modeled C inputs required to reach the target depended linearly on the initial SOC stocks, raising concern on the feasibility of the 4 per 1000 target in soils with a higher potential contribution on C sequestration, that is soils with high SOC stocks. Our work highlights the challenge of increasing SOC stocks at large scale and in a future with warmer climate."**

Specific comments:

1. The introduction is interesting, but sometimes strays from the subject of the study.

78-89: This paragraph is too detailed and could be halved in length, especially given it is not the focus of the paper.

**We removed L. 84 to L.89 and cited Chenu et al. (2019), who already developed these concepts.**

90-108: Again, this paragraph is too detailed and could be halved in length, given it is not the focus of the paper.

**We removed L.95-L.102.**

L109: In contrary, this paragraph is the focus of the paper and needs to be expanded. E.g. the concepts of feasibility and applicability of the 4p1000 need to be introduced; is there doubt that the 4p1000 cannot be reached / maintained ? Why? Are there indications that there is not enough available biomass to reach the 4p1000 aim? State that this study focuses on C inputs only and not on reducing C loss (decreasing mineralisation through cover crops, otherwise reducing erosion).

At the moment there is little justification why this study is necessary.

**We rephrased the part stating the objectives, better introducing the context of this work, as follows:**

**"The feasibility and applicability of a 4‰ increase target depend on biotechnical and socio-economic factors. As we mentioned earlier, a number of practices are known to increase SOC stocks in agricultural systems. However, it is still debated whether they will be sufficient to reach the 4p1000 objective. Minasny et al. (2017) described opportunities and limitations of a 4‰ SOC increase in 20 regions across the world. Several authors (e.g. Baveye et al., 2018; van Groenigen et al., 2017; VandenBygaart, 2018) argued that some of the examples described in Minasny et al. (2017) were not representative of wide-scale agriculture and suggested that a 4‰ rate is not attainable in many practical situations (Poulton et al., 2018). Implementing new agricultural practices that allow the maintenance and increase of SOC stocks might require structural land management changes that not all farmers will be willing to adopt. Incentivizing and sustaining virtuous practices to increase SOC stocks should be a strategy for policymakers to overcome socio-economic barriers (e.g. Lal, 2018; Soussana, 2017) and in order to do that, they need to be correctly informed. Recent works have assessed the biotechnical limitations of a SOC increase, studying the required and available biomass to reach a 4p1000 target in European soils (Wiesmeier et al., 2016; Martin et al., 2021; Riggers et al., 2021).**

**Our work was set up in this context with the objectives to: 1) estimate the amount of C inputs needed to increase SOC stocks by 4‰ per year; 2) investigate if this amount is attainable with currently implemented soil practices (i.e. organic amendments and different crop rotations) and 3) study how the required C inputs are going to evolve in a future driven by climate change. We used the biogeochemistry SOC model Century, which is one of the most widely used and validated models (Smith et al., 1997) to simulate SOC stocks in 14 different agricultural LTEs around Europe. We set the target of SOC stocks increase to 4‰ per year for 30 years, relative to the initial stocks in the reference treatments. With an inverse modeling approach, we estimated the amount of additional C inputs required to reach a 4p1000 target at these sites. Finally, we evaluated the dependency of the required additional C inputs to different scenarios of increased temperature."**

2. The methods are generally clearly described, with one exception: I have only a basic understanding of the CENTURY model, but as far as I understand from the manual and from Parton et al. (1993), plant C inputs into the soil are based on plant production, as simulated by the model. The authors however calculated plant C inputs based on allometric equations (the approach first described by Bolinder), using user-given yield data as inputs. Is this an alternative version of the CENTURY model? If so, this needs to be explained.

**Yes, we used a version of Century that only simulates SOC, we add this information in the description of Century model as follows:**

**"In the version used, only SOC is modeled and plant growth is directly accounted as variations of C inputs."**

3. This study asks how much C inputs are needed to reach the 4p1000 target i.e. in the future. Because recent climate data were used for the simulations, the result has probably been underestimated, as SOC decomposition rates in a warmer world might be higher than at current. Indeed, this has been demonstrated by similar studies (e.g. Riggers et al 2021, 10.1007/s11104-020-04806-8; Wiesmeier et al. 2016, 10.1038/srep32525). The authors are aware of this and addressed the issue with the so-called 'sensitivity analysis'. This is good, but the outcome of this analysis needs to be related more closely to the main results, especially in the abstract. I therefore suggest that the sentence of L50-51 be moved to the end of the previous paragraph in the abstract, as follows: At the end of the sentence on L 48, add: "This means that the C inputs required to reach the 4p1000 target might actually be much higher. Furthermore, we estimate that annual C inputs will have to increase even more due to climate warming, that is 54% more and 120% more, for a 1°C and 5°C warming, respectively."

Furthermore, I recommend the two studies mentioned above be cited in the discussion.

**The sentence was added, as indicated. We added the citations of Wiesmeier et al. 2016 and Riggers et al 2021 both in the introduction and in the discussion:**

**L.115: "Recent works have assessed the biotechnical limitations of a SOC increase, studying the required and available biomass to reach a 4p1000 target in European soils (Wiesmeier et al., 2016; Martin et al., 2021; Riggers et al., 2021)."**

**L.491: "Riggers et al. (2021) found that in 2095, a minimum increase of C inputs by 45% will be required to maintain SOC stocks of German croplands at the level of 2014. However, they found that to increase SOC stocks by 4‰ per year, a much higher effort will be required. That is, C inputs in 2095 will have to increase by 213% relative to current levels."**

**L. 595 "Moreover, inaccuracies in simulations outcomes, such as those found in this study, need to be reduced. As discussed in subsection 4.2.3, a better representation of C inputs dynamics and management practices could improve the simulation of SOC stocks.**
**We suggest to consider multi-model analysis for this type of work in the future (Farina et al., 2021), to acknowledge different representations of SOC and reduce the effect of single models' uncertainties. Furthermore, the likely increase of SOC mineralization due to future climate change (Wiesmeier et al., 2016) needs to be taken into account."**

4. L361-363: Isn't this increase in temperature an overestimate? The 1 and 5 degree increases refer to increases between 1996 (roughly half way between 1986 and 2005) and 2090 (roughly half way between 2081 and 2100), i.e. 94 years. The simulations for the study are however run only for 30 years, meaning such temperature increases should not be expected to occur.

**We agree that a +1°C and +5°C are likely overestimations of the temperature increase for a 30 years simulation period. However, our objective was not to apply climate change scenarios, rather understanding the sensitivity of the model to temperature. To better show this point, we reformulate the following points:**

- **We change the "Sensitivity analysis to temperature" section (2.3.3), as follows:**

"We tested the sensitivity of model outputs to temperature, running two simulations with increased temperatures. We considered two representative concentration pathways (RCPs) of global average surface temperature change projections (IPCC, 2015). The first scenario (RCP2.6) is the one that contemplates stringent mitigation policies and predicts that average global land temperature will increase by 1°C during the period 2081-2100, compared to the mean temperature of 1986-2005. The second scenario (RCP8.5) estimates an average temperature increase of +4.8°C, compared to the same period of time. We ran two simulations of increasing temperature scenarios with Century. We considered the same initial conditions as the standard simulations, hence running the spin-up with the average soil temperature and relative humidity of the 30 years preceding the experiments. Then, we increased daily temperature by 1°C (AS1) and 5°C (AS5) for the entire simulation length, to assess the sensitivity of modeled C inputs to increasing temperatures. Nevertheless, it must be noted that our simulations are running over a 30 years period not the entire 21st century. Thus, the temperature sensitivity analysis should not be considered as a test of climatic scenarios but like a classical sensitivity analysis where the boundaries were defined following RCP2.6 and RCP8.5 predictions of increased temperatures."

- **We changed section 3.3 title to: "Carbon input requirements with temperature increase"**

- **We added a sentence in the sensitivity analysis discussion, stating that:**

"Although these scenarios (+1°C and 5°C) are calculated over ~100 years, we used these values over a 30 years' simulation to assess the sensitivity of Century to temperature increase."

5. L496: "farmers may have difficulties in producing or buying high quantities of EOMs" Surely this is not the point: Even if farmers are able to source additional EOMs, these EOMs will then be lacking from other sites (from whence they came), meaning that there is no net removal of CO2 from the atmosphere -> no C-sequestration. Only if these EOMs would otherwise be mineralised (e.g. burnt) can these be considered sequestration.

We rephrase our sentence to take this into account:

"First of all, the amount of organic fertilizers is limited at regional scale. If farmers source additional EOMs elsewhere, only those EOMs that otherwise would be mineralized (e.g. burnt) and not applied to land account as sequestration. Second, farmers may be prevented from applying high amounts of EOM because of the risk of nitrate and phosphate pollution (Li et al., 2017; Piovesan et al., 2009). Moreover, producing additional animal manure

**implies larger GHG emissions through animal digestion and manure decomposition. Consequently, even if more manure is returned to the soil, it will not necessarily result in climate change mitigation."**

6. L632-633: "indicating that lower amounts of carbon inputs might be sufficient to reach the 4p1000 target where SOC stocks are low." Is this effect (1) merely a calculation effect? i.e. Soils with high SOC stocks need more to reach the 4p1000 target simply because 0.4 % of a higher value is higher (than 0.4 % of a lower value); or (2) is there additionally a 'true' effect e.g. the application of 1 t C / ha of EOMs to soils with lower SOC stocks results in a higher SOC gain than 1 t C / ha of EOMs to soils with high SOC stocks?

If the answer is (2), this means that the addition of EOMs to soils with low SOC is much more efficient. This is very important, given that EOMs are a finite resource. In this case, I would also suggest rephrasing the sentence. A suggestion, "indicating that the 4p1000 target can be more efficiently met, i.e. using fewer EOMs, in soils with lower SOC stocks" or "indicating that more C can be stored, and the 4p1000 target more efficiently met, i.e. using fewer EOMs, in soils with lower SOC stocks"

**We believe this is mainly a modeling effect, because a similar trend was not found in field treatments. However, it is hard to make conclusions in this sense, because only one site had very high initial SOC stocks.**

Technical comments:

L30-31: I recommend removing «to promote better agricultural practices». Firstly, it distracts from the focus of the paper and secondly, this is strictly speaking not the aim of the initiative but rather how the aim can be achieved.

**We removed "to promote better agricultural practices".**

L32: The term "straightforward" is ambiguous. Please be more precise.

**We changed this term and rephrased the sentence saying: "One way to enhance SOC stocks is to increase C inputs to the soil."**

L36-37: "Initial simulated stocks were computed analytically assuming steady state". I think "initial stocks were simulated assuming steady state" would suffice. Alternatively, consider removing the whole sentence (too much detail for the abstract?).

**We rephrased this sentence as suggested.**

L40: control plot -> control plots (otherwise it is implied the model was calibrated to the control plot of a single experiment)

**We specified "control plots".**

L 40-41: "conventional management without additional carbon inputs" Additional to what? Is the term 'no exogenous carbon inputs' meant here? The authors need to be more precise, because in some countries, conventional management does include adding EOMs, e.g. farmyard manure.

**We specified: "conventional management without additional carbon inputs from exogenous organic matter or changes in crop rotations"**

L47-48: why "on the variation of SOC stocks"? Is not simply "the SOC stocks" meant here? I assume that the SOC stocks are modelled as increasing by 0.25 %? Please specify.

**We rephrased as follows: "However, Century might be overestimating the effect of additional C inputs on SOC stocks. In the experimental sites, we found that SOC stocks in treatments with additional carbon inputs were increasing by 0.25% on average."**

L56: "Strategies of conservation and expansion of existing SOC pools may be necessary but not sufficient to mitigate climate change" The word "are" needs to be inserted before "not sufficient".

**The word "are" was inserted before "not sufficient"**

L68: "resilience face to changes in climate" -> resilience to changes in the climate.

**We changed "resilience face to changes in climate" to "resilience to changes in the climate".**

L71: assessed -> demonstrated (otherwise, depletion of SOC could have been assessed, but have been found to not be the case).

**We changed the word "assessed" to "demonstrated"**

L74: "carbon in the soil" -> "C in the soil" or simply "SOC"

**We changed "carbon in the soil" to "SOC"**

L78: The word 'balance' is inappropriate (they could be unbalanced and SOC stocks would still be present). A suggestion: "SOC stocks are a function of C inputs and C outputs."

**We changed "balance" to "SOC stocks are a function of C inputs and C outputs."**

L86-89: I understand what the authors mean to say, but fear that this sentence is unclear for readers unacquainted with the literature / concept of C-sequestration. I suggest inserting 'at a given location' between "SOC stocks over time" and "and is not necessarily". It would also be useful to give an example "for example, EOM added to one site is not an example of C-sequestration if it results in loss of EOM at another site".

**This sentence was entirely removed to take into account comment #1.**

L90: "for estimating" -> to estimate and "evaluating" -> evaluate

**We changed "for estimating" to "to estimate" and "evaluating" to "evaluate".**

L96: kept on -> maintained L98: In fact, -> For example,

**Both sentences were removed to take into account comment #2.**

L104: allow estimating the evolution of SOC stocks and their future trends to assess the potential gain of SOC at global scale -> allow the evolution of SOC stocks and their future trends to be estimated, enabling the potential gain of SOC at a global scale to be assessed, also following changes in agricultural practices.

**We rephrased the whole sentence to take into account this comment and comment #2, as follows:**

**"Combining measurements of SOC with models provides a wider applicability of the information collected in field trials, as it allows SOC stocks and their future trends to be estimated."**

L127: one control plot -> one control plot in each experiment.

**We changed "one control plot" to "one control plot in each experiment".**

L131: "experiments with a duration of at least 10 years," Surely this is redundant if the experiments all ran for at least 11 years? If this phrase is necessary, please explain what is meant here.

**We removed "with a duration of at least 10 years"**

132: except from Foggia -> except for Foggia. It needs to be made clear here that control plots also receive no EOM. I suggest: "C inputs in all sites except for control plots, and all plots in Foggia included..."

**We rephrased as suggested: "C inputs in all sites except for control plots, and all plots in Foggia included exogenous organic matter (EOM) addition"**

L138: remove 'found'

**We removed "found"**

Table 1: Please specify whether 'N' represents the addition of nitrogen or of mineral fertiliser (obviously including nitrogen).

**It is specified in the bottom notes: "**Optimal amounts of mineral fertilizers added to the control plot and to all other treatments in the experiment"**

L151: a part -> apart

**We corrected a part -> apart**

L154: ratio: should this not be 'rate'?

**We corrected ratio -> rate**

153-157: I presume the values here all pertain to Table 1. If so, can the authors please recount: the I get only 18 (including the 0.40 score in Arazuni) treatment cases with a SOC annual variation of > 0.4 %. I counted 6 cases where the increase is less than 0.4 % and 22 times where there is decrease in SOC stocks.

**We recounted and corrected these values, as suggested.**

Figure 1 is not referenced in the text. Please add where appropriate.

**Referenced at L.126**

L167: Why not give a range for humidity (as for temperature)? This is more useful to know.

**We added the range for humidity.**

L173: Table legend should begin with: "Information about experimental sites" (or something like this).

**We changed the table legend as follows: "Information about experimental sites, including: mean annual temperature (C°) and soil humidity to approximately 20 cm depth ($kg_{H2O}$ m$^{-2}$$_{soil}$) simulated with the ORCHIDEE model at each experimental site, measured pH, bulk density (g cm-3), clay (%) and initial SOC stocks in the control plots (MgC ha$^{-1}$) at the experimental sites. Reference papers for each site are indicated."**

L197: Please define the term 'SOC(t)' in text

**We added "and SOC evolution with time (SOC(t))"**

L199: space required between "(t)" and "of"

**Space added**

L235: consists in -> consists of

**We changed "in" to "of"**

L243: "By enhancing the computational performance of the simulations". What does this mean? Please be more precise. Or omit.

**We changed "By enhancing the computational performance of the simulations" to "By speeding up the performance of the simulations"**

L244: "this technique enables the analysis of system properties and facilitates studying model behavior." I think this phrase is redundant. Omit? (in which case add "also" between "It and "allowed" in the next sentence.)

**We removed this sentence.**

L346: please specify what "s is the size of the experiment" means. This is unclear to me.

**We changed "size" to "length"**

L369: "fitted quite well the observed SOC stocks" -> fitted the observed SOC stocks well

**We changed "fitted quite well the observed SOC stocks" -> fitted the observed SOC stocks well**

L370: remove comma in "Fig 4.a, provides"

**We removed comma**

Figure 4: The colour scheme is nice, but colours need to be more saturated as they are difficult to distinguish from one another if the computer screen is dim.

**The colours of the picture were changed to make it more saturated**

L383: Could the term 'optimized' be specified here ? e.g. minimum C input additions needed / required to reach the 4p1000 target? Both terms are used elsewhere (e.g. L 422, L451) and make more sense than 'optimized'.

**The term "optimized" was changed into "required"**

L389: why 'globally'? Is this word necessary here? L394: per year) -> per year respectively)

**We changed the term "globally" into "on average among the studied sites"**

L399-400: The absolute increases of the active and passive pool need to be given (so that the 2.7 MgCha-1 can be compared to something).

**This info was added: "(against an increase of 0.1 and 0.06 MgC ha$^{-1}$ in the active and passive compartments, respectively)".**

L410-412: This sentence belongs in the discussion.

**This sentence was moved in the discussion, to L.578**

L414: How can it be that the TOT C pool requires a lower C input change than all the other pools? I would have expected this pool to be a weighted average of the other pools and thus have an intermediate value.

**TOT C is the average total carbon input change among the sites. It was not calculated as the weighted average of the other pools, but as the percentage change between total initial C inputs and "final" C inputs (i.e. C inputs after the 4p1000 optimization)**

**We add a sentence in Figure 5 caption to clear this out: "N.B: Total change of carbon inputs (TOT) was calculated as the percentage change between the amount of carbon inputs before and after the 4p1000 optimization, averaged across all sites."**

L452: "increase on average by 54%": please specify what this is compared to. Compared to the C inputs required to meet 4p1000 (CURR) or compared to the control plots of CURR (I

**We add ", relative to current C inputs in the control plots"**

L454: "business as usual situation" is this referring to the CURR climate? This is stated in the legend to Fig. 9 but also needs to be stated in the main text.

**We add "(CURR)" to the end of the sentence in the main text**

Figure 9: The y-axis label needs to be changed; at present it implies the average C input change required to reach 4p1000 is a decrease in C inputs (circa 50 %)

C inputs change (%) -> C inputs increase (%)

**The y-axis label was changed: C inputs change (%) > C inputs increase (%)**

L466-468: This allowed us: 1) taking into account the average carbon quality of the litter pools in the different crops rotations and 2) correctly estimating the initial values of SOC stocks on the majority of the sites. -> This allowed us to: 1) take into account.... and 2) estimate correctly....

**We corrected this sentence as suggested.**

L468: On the other side -> On the other hand

**We changed the term "side" to "hand"**

L469: why 'redistribution' ? Is simply 'distribution' meant here?

**"Redistribution" is meant here because it refers to the allocation of C in the different pools, after the 4p1000 optimization. Hence, while there might be some initial predicted distribution, we refer here to the redistribution after the optimization.**

L468-469: If this is mentioned here, it needs to be discussed in more detail. For example, is there any evidence that this impacted the main outcomes of this study, e.g. the amounts of C inputs required?

**We added figure C2 in the appendix C (see Fig_C2.pdf in the supplement) to show that the optimization does not affect model outputs for current values of temperature in our modeling exercise. We added a few sentences to discuss the optimization effect on modeling results: "Figure C2 shows that the optimization of temperature sensitive parameters did not affect significantly the required C input estimation for the current temperature scenario. This means that, although parameter optimization improved the simulation of SOC stocks in the control plots, the final results are not affected by it."**

[Figure]

**[Figure C2: Effect of the optimization of the $Q_{10}$ and reference temperature ($T_{ref}$) parameters on the additional carbon inputs to reach the 4p1000 predicted by Century (mean $\pm$ standard deviation).]**

L470: Initializing -> initialize L471: on -> regarding

**These two terms were corrected**

L472: we decided to optimize -> we optimized

**We changed "we decided to optimize" to "we optimized"**

L478-479: It is unclear what is meant by "simulate SOC stocks variation". Is simply "simulate SOC stocks" meant here?

**We removed "variation"**

L478-483

L502: different whether -> different depending on whether

**We added "depending on"**

L507: "The additional carbon is essentially slow" Carbon is not 'slow'. Please rewrite more precisely.

**We changed "is essentially slow" to "is mainly stored in the slow pool"**

L510-511: we might consider implementing new strategies of additional carbon later on -> new strategies of additional C could be implemented later on.

**This sentence was corrected, as suggested.**

L512: residues -> residue

**This term was corrected**

L535: I recommend a sentence such as "Thus, Century seems to be over-predicting the effect of adding C inputs in the virtual simulations" after the word "experiments." It would make the paragraph much easier to read.

**This sentence was added.**

L540: in -> for

**This term was corrected**

L542-543: What is meant by this sentence ("The calculation method...")? Please expand.

**We rephrased this sentence as follows: "The method used to estimate C inputs (i.e. the allometric functions from Bolinder et al. (2007) in our case) also influences the simulation of SOC stocks"**

L543: relatively -> relative

**This term was corrected**

L546: A value of N inputs is given. Why is the term 'proportion' used to describe this amount (in the line above)?

**We changed the term "proportion" into "amount"**

L552: inputs -> input

**This term was corrected**

L552: I recommend: need -> requirement ('need' is correct but because this is also a verb, the is sentence difficult to understand)

**The term "need" was substituted with the term "required"**

L557-571: This section of text is not very focussed and I did not understand what the authors are trying to say. The values cited are important for the discussion of EOMs and agricultural soils, but how does this text relate to the rest of the paragraph?

**We rephrased this part to better link it to the rest of the paragraph:**

**"Pellegrini et al. (2016) reported the amounts of sewage sludge disposed on landfill in Europe (EU26) from Eurostat (2014b). In 2010, this was 0.914 Tg DM. Using the Van Bemmelen factor (1.724) to convert OM to OC (McBratney and Minasny, 2010; Rovira et al., 2015), we estimated that the sewage sludge disposed on landfill in Europe was around 0.004 MgC ha$^{-1}$ per year in 2010. If applied to cropland, this could potentially increase C inputs to the soil and decrease GHG emissions associated to landfilled waste. However, in some countries social acceptability of spreading EOM such as sewage sludge is very low, limiting its actual potential. In Europe, landfilled municipal waste was 0.3 MgC ha$^{-1}$ in 2019 (estimated from Eurostat (2020) considering a C content in household waste of 71% (Larsen et al., 2013)). This is higher than the amount of municipal waste currently composted in Europe (i.e. 0.22 MgC ha$^{-1}$ in 2019, according to Eurostat (2020)), showing that additional efforts to improve the reutilization of municipal waste could help to increase C inputs in agriculture."**

L573: Is the term 'decreases' meant here instead of 'increases? Please check.

**Yes, we changed the term "increases" to "decreases"**

L577: As we could expect -> As expected (or 'As we expected')

**We changed "As we could expect" to 'As we expected'**

L593-595: This sentence is important, but it needs to be expanded. Most importantly,

inaccuracies / biases in the outcomes of simulations – such as that found in this study – need to be reduced. The authors could incorporate suggestions from L535-538 (e.g. we need to represent the dynamics / timing of C inputs more accurately). Furthermore, the (probable) additional reduction of SOC due to future climate needs to be taken into account.

**We rephrased and expanded subsection 4.2.5 (cf comment below)**

L594-545: The term 'cut out model uncertainty' is awkward. Multi-model analysis does not strictly speaking omit model uncertainty, but rather acknowledges it and reduces the effect of extreme model outcomes. Please reword.

**We take into account both comments by rephrasing and expanding subsection 4.2.5 as follows:**

**"Also, inaccuracies in simulations outcomes – such as those found in this study – need to be reduced. As discussed in subsection 4.2.3, a better representation of C inputs dynamics and management practices could improve the simulation of SOC stocks. We suggest to consider multi-model analysis for this type of works in the future, to acknowledge different representations of SOC and reduce the effect of single models' uncertainties. Furthermore, the likely increase of SOC mineralization due to future climate change (Wiesmeier et al., 2016) needs to be taken into account."**

L601-602: This sentence lacks a context and needs to be written more precisely. I presume the authors mean that SOM will increase decomposition under future (warmer) climates because its decomposition rate is affected – in general increased – with increasing temperatures.

**We rephrased this sentence as follows:**

**"We know that the decomposition rate of SOM is affected – in general increased – with increasing temperatures."**

L602-603: remove 'from SOC stocks' L603: of -> by

**We removed 'from SOC stocks'**

**We changed "of" into "by"**

L603-604: This sentence needs to be explained. I presume the authors would like to say that there is a diversity of compounds and therefore a diversity of changes that can be expected. As well as the diversity of strength of sensitivity of decomposition rates to future climates, there might also be increases as well as decreases in decomposition rates in the future (e.g. water limitation will reduce some decomposition rates).

**We expanded this sentence with the following explanation:**

**"In fact, a diversity of responses of decomposition rates to future climates can be expected, including increases due to higher temperature as well as decreases due to water limitation**

L613: inputs -> C inputs

**We added "C"**

L619: The Davidson and Janssens paper addresses this. If there is space, I recommend to cite this paper again here, and briefly mention how moisture change is important.

**We added the following: ", with consequences on root respiration and microbial decomposition (Davidson and Janssens, 2006)."**

L627-628: I would recommend removing this sentence. It is not a generalizable finding but rather is how the authors came to the conclusion that Century over-estimated the effects of the C inputs.

**We consider this sentence important because it compares current practices to model simulations of additional C inputs, which was one of the objectives of this study. This finding (see Fig. 8) does not show that Century over-estimates the effects of C inputs on SOC stocks. It rather shows that current practices use higher amount of C inputs than those predicted by Century. In a further analysis we showed and discussed that the model might over-estimate the effects of C inputs, hence additional C inputs to reach the 4p1000 might be actually higher than those predicted. This was pointed out in the results, discussion and conclusion parts.**

L628-629: might be overestimating -> might have overestimated L635: The required amount -> Furthermore, the required amount L636: to substantially increase -> to increase substantially L636: rising concern on –> raising concern about

L637: SOC stocks -> SOC stock

**These sentences were corrected**

L638-640: The matter covered in this sentence is not a conclusion reached from this study and it does not fit into the rest of the conclusion. I recommend removing it.

**This sentence was removed**

L641: largely depends -> depends largely L642: only considered -> considered only L644: is worthwhile to -> should

**These sentences were corrected**

L645: "to allow for a correct estimation of the uncertainties related to model-specific assumptions" This is too simplistic. Though it would allow the influence of extreme model outcomes to be reduced, if all simulations are biased (e.g. over-estimate the effects of C inputs) then the results of the multi model analysis will also be biased. A more urgent research priority would be to address the cause(es) of the bias(es). See above (response to L593-595). Additionally, EOMs need to be accurately represented in models. The decomposition of biochar for example – the use of which might become more widespread in the future – has very different decomposition dynamics than other EOMs and models need to be adjusted to account for this.

**We replace this sentence with the following:**

**"Causes of biases in model simulations should be addressed in future studies and the representation of C inputs should be improved. We also suggest that future research should include multiple models to reduce the influence of extreme model outcomes on the representation of SOC stocks."**

**General technical comments:**

- The authors use the terms 'site' and 'experiment' interchangeably. I presume these terms relate to the same thing. If so, please use one term only throughout the manuscript (I would suggest 'site').

**We changed "experiment" to "site" throughout the manuscript.**

- Likewise the terms control plots and reference plots (e.g. L473 and in table 1)

**We changed "reference plot" to "control plot" throughout the manuscript.**

- The abbreviation 'SOC' is not used everywhere (e.g. L 418) but should be.
- Likewise, 'C' (carbon).

**We corrected the abbreviations "SOC" and "C" throughout the manuscript.**

**Referee 3**

**Summary**

Bruni and co-authors synthesize a wealth of observational data from cropping experiments and explore in changes in C inputs that would be needed to meet 4 per mil goals across European agricultural sites with the CENTURY model. The paper is well written and appropriate for the journal. Before final publication, however, greater attention needs to be given to model validation to contextualize the projections being made.

**We thank the reviewer for the positive comment. We added two figures (see Fig_5.pdf and Fig_C2.pdf in supplement) to the manuscript, to better show the validation of model results. Also, we added some explanations and comments to better contextualize model projections.**

**Major concerns**

I assume these papers follow a rather prescriptive formulation that involves model calibration, validation, and implementation / implication sections.

The model calibration is presented (Table 3, Fig 4), where the authors calibrate the litter quality, Q10 and reference temperature for control plots at the 14 sites. The findings raise some concerns, described below, but the discussion generally handles this well.

I would assume that model validation would involve projecting trends in soil C stocks from the modified C input experiments. I appreciate that the observations of trends from these experimental treatments are highly variable (line 155 and Table 1). But there is seemingly no effort to validate the ability of the model to capture the correct magnitude of soil C change (relative to controls) in the additional carbon experiments that are listed. Revisions to the manuscript should more clearly illustrate how well the calibrated CENTURY model captures observed soil C changes across these experimental plots in response to experimental manipulations.

**We added Figure 5 to show Century predictions of additional C inputs to reach the 4p1000 target, together with additional C inputs experiments and their relative SOC stock variation. We show that Century actually overestimates the effect of additional C inputs on SOC stocks. However, treatments effect on SOC are highly variable.**

**We add the following sentence to subsection 3.1:**

**"We tested the capability of Century to reproduce SOC stocks increase in the additional C input treatments (Fig. 5). Figure 5 shows the correlation between additional C inputs and SOC stock increase in the C input treatments ($R^2 = 0.23$). In the same graph, we can appreciate additional C inputs simulated by Century to reach the 4p1000 target being 0.66 $\pm$ 0.23 MgC ha$^{-1}$ per year (mean $\pm$ standard deviation from the mean). This shows that Century is generally overestimating the effect of additional C inputs on SOC stocks increase. However, the effect of additional C inputs on observed SOC stock increase varies largely across different treatments."**

**We suggest that this overestimation might be due to the hypothesis of equilibrium and we add a discussion to subsection 4.1:**

**"The capability of Century to simulate SOC stocks in the simulations of additional C treatments might be a major shortcoming of modeling results. In fact, although SOC stocks were found to be increasing on average in the additional C treatments (0.25% per year with 1.52 MgC ha$^{-1}$ yearly additional C inputs), this increase rate is lower than the 0.4% increase of SOC stocks predicted by Century with lower amounts of virtual C inputs (0.66 MgC ha$^{-1}$ per year). This is pointed out in Fig. 5, where we can see that predicted additional C inputs to reach the 4‰ are lower than the correlation line between additional C inputs and SOC stocks increase in field treatments. The overestimation of the C input effect on SOC stocks in Century might be related to the assumption that SOC stocks are in equilibrium with C inputs at the onset of the experiment and on the high sensitivity of the model to C inputs."**

[Figure]

**[Figure 5: Correlation between additional carbon inputs (MgC ha⁻¹ per year) and annual SOC stock increase (%) in the carbon inputs treatments and mean ± standard deviation of the additional carbon inputs to reach the 0.4% target in Century.]**

Model implications are well documented and described.

I don't completely understand the details of the model calibration and optimization that were taken, but it's surprising to me that the metabolic:structural partitioning varies to such a great extent among experimental sites (Table 3). Notably, the aboveground M:S calibration largely flips between the upper and lower limit of the range allowed by the model (85:15 or 15:85). Does this seem realistic? I would assume there are general estimates for lignin:N ratios of different plant parts for different crops that could be used to help constrain this parameter?

**In Century, the standard function to calculate the metabolic part of the litter (LF$_m$) is the following:**

$$LF_m = \max\left(0.15, 0.85 - 0.018 \cdot \frac{L}{N}\right)$$

which ranges between 0.15 and 0.85 and where $\frac{L}{N}$ is the lignin to nitrogen fraction of the crop. The structural fraction ($LF_f$) is then calculated as:

$$LF_f = (1 - LF_m)$$

We researched L:N values in the literature and we found that:

- For winter wheat L:N = 42 (Adair et al., 2008) -> $LF_m = max(0.15, 0.094) = 0.15$
- For pea L:N = 1.149 (Ghimire et al., 2017) -> $LF_m = 0.83$
- For winter rapeseed L:N = 0.625 (Ghimire et al., 2017) -> $LF_m = 0.84$
- For oats L:N = 1.12 (Ghimire et al., 2017) -> $LF_m = 0.83$

These values are in line with those found by Century's optimization of the metabolic:structural ratio. We were not able to find L:N values for all crops, nor for different parts of the plant. For these reasons, we considered that model parameters' optimization was the best option for our simulations. Limits of this approach are discussed in subsection 4.1.

Similarly, why is there such a large range in the Q10 values for each site (range 2-5!). I understand this was done to help calibrate the model, but I wonder if the propensity for the calibration to settle on the extreme values for parameter ranges for litter quality and temperature sensitivity points to either: (a) structural deficiencies in the model or (b) equifinality issues from trying to fit a complicated model with sparse data. This is briefly discussed at the beginning of the discussion, but I wonder if it's a finding that should be noted in the abstract & conclusion as well?

Many authors found that Q10 values were highly variable across different pedo-climatic conditions (Craine et al., 2010; Lefèvre et al., 2014; Meyer et al., 2018; Wang et al., 2010). We decided to bind the model with the Q10 range found by Wang et al. (2010) in temperate regions. We agree that the optimization tends to settle the parameters on extreme values. It is difficult to conclude whether it is a structural limitation of the model or an equifinality issue. We rather suppose it is a combination of both.

We added figure C2 to appendix C, to show that the optimization does not affect model outputs for current values of temperature in our modeling exercise. In regard to this, we do not find it necessary to discuss optimization issues in the abstract or conclusion. However, we add some sentences in the discussion to clear out this point.

In subsection 4.1, we added:

"Figure C2 shows that the optimization of temperature related parameters did not affect significantly the required C inputs estimation, for the current temperature scenario. This means that, although parameters optimization improved the simulation of SOC stocks in the control plots, the final results are not affected by it."

[Figure]

**[Figure C2: Effect of the optimization of the $Q_{10}$ and reference temperature ($T_{ref}$) parameters on the additional carbon inputs to reach the 4p1000 predicted by Century (mean $\pm$ standard deviation).]**

To this point, the last sentence of section 4.1 is alarming, and suggest that CENTURY projects 2x the amount of SOC accumulation with 0.5x of the additional inputs, compared to observations. This raises two issues:

Lack of validation for the additional carbon experiments. If agricultural management practices or negative emissions strategies are supposed to be informed by studies this like (as implied in the introduction) it seems like the model projections may be overly optimistic by a factor of four! This raises some serious concerns that are somewhat glossed over by reading the abstract (and conclusion).

**As explained above, we added two figures to better show the capability of Century to reproduce SOC stocks in the additional C input treatments and discussed these issues in the results and discussion.**

**We also changed the last sentences of the abstract to stress out this point:**

**"This means that the C inputs required to reach the 4 per 1000 target might actually be much higher. Furthermore, we estimated that annual C inputs will have to increase even more due to climate warming, that is 54% more and 120% more, for a 1°C and 5°C warming, respectively. We showed that modeled C inputs required to reach the target depended linearly on the initial SOC stocks, raising concern on the feasibility of the 4 per 1000 target in soils with a higher potential contribution on C sequestration, that is soils with high SOC**

**stocks. Our work highlights the challenge of increasing SOC stocks at large scale and in a future with warmer climate."**

Minor and technical concerns:

Line 48, maybe this sentence can be clarified to address some of the concerns above?

**We added the following sentence after L.48:**

**"This means that the C inputs required to reach the 4p1000 target might actually be much higher."**

Line 77: I don't know what it means to 'promote a virtuous C cycle' and suggest this phrase be removed

**We removed 'and eventually promote a virtuous C cycle'**

Section 2.2 I understand that CENTURY also has functions where soil pH determines turnover times and soil texture (specifically sand content) modifies the partitioning of C fluxes between pools (and to CO2). It doesn't look these were used here, which is fine, but it should be clarified in the text.

**In this study, we used the original version of the Century model (Parton et al., 1987). To our understanding, Century in its original version does not have any function including soil pH for the C module (Foereid et al., 2007). Soil texture does affect the decomposition rates, but only as a function of clay, which we took into account (see Appendix A). However, a function of sand was not present in this version (see Fig. 1 in Parton et al. (1987)).**

Figure 4 is fine, but I guess I expected to see some kind of predicted vs. observed soil C stock plot as part of the model calibration?

**We added a figure of the fit between predicted and observed SOC stocks in the control plots to Figure 4.c (Fig_4.pdf in supplement)**

**And added the following sentence to the results:**

**"The correlation coefficient between modelled and observed SOC stocks in the control plots was 0.96 (Fig. 4.c)."**

**[Figure 4: a) Decomposed mean squared deviation (MgC ha-1)2 in control plots for all sites. LC = Lack of Correlation, NU = Non-Unity slope and SB = Squared Bias. b) Normalized root squared deviation (%) in control plots for all sites c) Fit of predicted versus observed SOC stocks (MgC ha-1) in control plots for all sites (R2 = 0.96).]**

--

The word 'virtual' is used heavily throughout the text and especially in the results and discussion. We know that the simulated results are from a model. As such, I wonder the use of 'virtual' is potentially redundant (l e.g. "virtual simulations"; "virtual C inputs") and can be removed?

**We removed the term "virtual" throughout the text**

Line 564 should there be a +/- symbol here?

**There is a typo, it should be $4558 \times 10^3$ MgDM. We converted this value to TgDM, i.e. 0.4558 TgDM to make it consistent with other values in the text.**

**Referee 4**

Review of Additional carbon inputs to reach a 4 per 1000 objective in Europe: feasibility and projected impacts of climate change based on Century simulations of long-term arable experiments

This study used the Century model (well established mathematical model of plant-soil cycling to simulate soil carbon dynamic in different pools) to assessed the required level of carbon inputs increase to reach the 4 per 1000 target in 14 European long-term agricultural experiments. Originated on Minasny et al. (2017) paper, the 4p1000 initiative aims to promote better agricultural practices to maintain and increase soil organic carbon stocks with the principle that a 4‰ per year on average soil carbon increase for 30 years would partially mitigate anthropogenic greenhouse gas emission (in addition to improve soil fertility and food security). Using these 14 European long-term agricultural experiments, the authors compared Century computed carbon inputs to the actual carbon input from the different treatments used on the individual experimental sites. From the 14 sites, 13 had exogenous organic matter addition and only one had had different crop rotations as way of increasing carbon input. The exogenous input were: pig manure, cow manure, poultry manure, farmyard manure, green manure, sawdust, biowaste, sewage sludge, household waste. The Century model was calibrated to fit the control plot of the study sites. These control plots, which did not receive additional carbon inputs, were considered as conventional management (business as usual).

The analyses detected that Century model for the studied systems might be overestimating the effect of additional C inputs on the variation of SOC stocks in some sites. Overall, the authors found that (based on Century modelling), on average among the selected experimental sites, annual carbon inputs would have to increase on average by $43.15 \pm 5.05$ %, $(0.66 \pm 0.23$ MgC ha-1) per year, with respect to the control situation to reach the 4p1000 target. One very interesting feature of the study is that the authors analyzed how this would change under future scenarios of global warming (with modelling performed at simulated $+1C°$ and $+5C°$). Accordingly, it was estimated that annual carbon inputs would have to increase further due to temperature increase effect on decomposition rates, that is 54% and 120% for a 1°C and 5C° warming, respectively.

In my opinion, the study is original and has a high potential impact scientifically and for policy makers because it is a broad analysis about the very important issue of global warming mitigation. The study is in line with EGU Biosciences journal topics and the manuscript is well organized. The different sections are interesting to read and well written.

However, I consider that the paper in it current form has some important shortcomings not properly handled or discussed that could potentially produce misinterpretation for policy makers. In the introduction it is acknowledged that to use soil carbon stock to offset net annual $CO_2$ anthropogenic emissions to the atmosphere it is a all-encompassing increase that is needed. Specifically, line 60 states: "the [4p100] initiative ... that increasing global SOC stocks up to 0.4 m depth by 4p1000 (0.4%) per year ...". Conversely, the large majority of sites and treatments used in the analyses were with animal manures as source of extra added carbon to reach the 4p1000. This brings two fundamental issues. First, the use of animal manures is generally not considered an improved agri-management practice but a business as usual scenario that unlikely could significantly be expanded. Thus, arguably, these animal manure treatments would be a type of standard practice and in these farms using customary animal manure another additional carbon input would be needed to reach the 4per1000. Second, the exogenous organic matter likely created a leakage carbon stock effect. That is, the soil carbon balance of the farmlands where the animal feeds were cultivated, harvested and exported need to be taken into account. Specifically, these lands could be carbon negative which would counterbalance the carbon gain where the manure is applied. Also, to really offset net annual $CO_2$ anthropogenic emissions the complete life cycle to the carbon is needed which include emissions by the animals, in the processing, transport and application of the manure. It is likely impossible to incorporate all these aspect into a single scientific paper but this manuscript should clearly discuss these limitations and mention in the conclusion that the results does not represent the full carbon balance. Finally, the discussion section should recognize that other potential beneficial management practices were not included in the study (e.g. cover crops, reduced tillage, biochar application, increased irrigated areas, improved pH, landscape diversification, mineral amendments, etc.) and that further studies should try to find medium to long-term experiments with this managements to determine if they follow Century modelling predictions to increase soil carbon storage enough to achieve 4per1000.

Overall, I consider that the necessary addition to the manuscript is considerable. I suggest major revision.

I have no specific comments, as stated above, the manuscript is well written.

We thank the reviewer for the comment and we tackle the limitations raised by changing several parts of the manuscript. Please find below the details, separated into three main points.

1.

In the introduction (now rephrased to clear out our objectives) we stated that we estimated if a 4‰ SOC increase is attainable with currently implemented soil practices:

"Our work was set up in this context with the objectives to: 1) estimate the amount of C inputs needed to increase SOC stocks by 4‰ per year; 2) investigate if this amount is attainable with

currently implemented soil practices (i.e. organic amendments and different crop rotations) and 3) study how the required C inputs are going to evolve in a future driven by climate change."

Also, in the discussion (L.554 – 556) we said:

"In terms of C sequestration, organic fertilizers coming from animal manure are usually being applied to the soil at some location, hence they cannot account for additional climate mitigation potential (Poulton et al., 2018)."

We add the following sentence:

"Rather, they are considered as a business as usual situation that can unlikely be significantly expanded."

However, some treatments considered in this study were sewage sludge and household waste, which are currently not a business as usual scenario. We discussed this in Subsection 4.2.4:

"In our study, we also considered treatments with other types of EOM addition, such as sewage sludge and household waste. In many countries, a significant proportion of food and urban waste is currently left on disposal areas, where C is lost to the atmosphere as $CO_2$ or methane ($CH_4$) emissions (Bijaya et al. 2006). Pellegrini et al. (2016) reported the amounts of sewage sludge disposed on landfill in Europe (EU26) from Eurostat (2014b). In 2010, this was 0.914 TgDM. Using the Van Bemmelen factor (1.724) to convert OM to OC (McBratney and Minasny, 2010; Rovira et al., 2015), we estimated that the sewage sludge disposed on landfill in Europe was around 0.004 MgC ha$^{-1}$ per year in 2010. If applied to cropland, this could potentially increase C inputs to the soil and decrease GHG emissions associated to landfilled waste. However, in some countries social acceptability of spreading EOM such as sewage sludge is very low, limiting its actual potential. In Europe, landfilled municipal waste was 0.3 MgC ha$^{-1}$ in 2019 (estimated from Eurostat (2020) considering a C content in household waste of 71% (Larsen et al., 2013)). This is higher than the amount of municipal waste currently composted in Europe (i.e. 0.22 MgC ha$^{-1}$ in 2019, according to Eurostat (2020)), showing that additional efforts to improve the reutilization of municipal waste could help to increase C inputs in agriculture."

2.

We agree that sourcing EOM from another field does not account as additional C sequestration. Hence, we remove the following sentence from the discussion: "First of all, the amount of organic fertilizers is limited at site scale and farmers may have difficulties in producing or buying high quantities of EOMs (Poulton et al., 2018). "

Instead, we add the following:

"However, a full C cycle assessment should be considered to make sure that GHG emissions associated to such treatments do not exceed additional C storage (Guenet et al., 2020)."

Also, we add the following sentence to the end of subsection 4.2.1:

"Moreover, producing additional animal manure implies larger GHG emissions through animal digestion and manure decomposition. Consequently, even if more manure is returned to the soil, it will not necessarily result in climate change mitigation."

And to subsection 4.2.4:

"In this study, we did not include other potentially beneficial management practices, such as cover crops, reduced tillage, biochar application, improved soil pH, landscape differentiation and mineral amendments. Further research should investigate if long-term experiments with these management practices would be able to increase SOC stocks by 4p1000, following Century predictions."

3. In the conclusion, we add the following sentence

"In this study, we did not take into account the whole life cycle of C at the farm. However, compensating $CO_2$ emissions from human activities through SOC sequestration should also comprehend GHG emissions related to the management of additional EOM."